# Multi-Temporal Sentinel-2 Data Analysis for Smallholding Forest Cut Control

**Alberto López-Amoedo** [1], **Xana Álvarez** [2,*] , **Henrique Lorenzo** [3] and **Juan Luis Rodríguez** [3]

[1] Asefor Ingeniería Forestal, S.L.E. Centro de Emprendemento Monte Gaiás, Cidade da Cultura, 15707 Santiago de Compostela, Spain; berto@gestionforestal.es
[2] School of Forestry Engineering, University of Vigo, Campus A Xunqueira s/n, 36005 Pontevedra, Spain
[3] CINTECX, GeoTECH Research Group, Universidade de Vigo, 36310 Vigo, Spain; hlorenzo@uvigo.es (H.L.); jlsomoza@uvigo.es (J.L.R.)
* Correspondence: xaalvarez@uvigo.es

**Abstract:** Land fragmentation and small plots are the main features of the rural environment of Galicia (NW Spain). Smallholding limits land use management, representing a drawback in local forest planning. This study analyzes the potential use of multitemporal Sentinel-2 images to detect and control forest cuts in very small pine and eucalyptus plots located in southern Galicia. The proposed approach is based on the analysis of Sentinel-2 NDVI time series in 4231 plots smaller than 3 ha (average 0.46 ha). The methodology allowed us to detect cuts, allocate cut dates and quantify plot areas due to different cutting cycles in an uneven-aged stand. An accuracy of approximately 95% was achieved when the whole plot was cut, with an 81% accuracy for partial cuts. The main difficulty in detecting and dating cuts was related to cloud cover, which affected the multitemporal analysis. In conclusion, the proposed methodology provides an accurate estimation of cutting date and area, helping to improve the monitoring system in sustainable forest certifications to ensure compliance with forest management plans.

**Keywords:** remote sensing; forest cover; *Eucalyptus globulus*; *Pinus pinaster*; time series

## 1. Introduction

Forest ecosystems provide multiple ecological, economic and social benefits. They harbor a portion of the world's biodiversity, play a key role in regulating water flows, protect soils and contribute directly to national incomes and the local livelihoods of millions of people worldwide [1]. Furthermore, these ecosystems regulate key aspects of the global carbon cycle [2] and weather patterns via a number of different mechanisms, such as forest albedos [3], sensible heat and aerodynamic roughness [4,5]. Specifically, forest areas are among the land use types that capture and store more carbon than they use. For example, returning farmland to larch forests has been shown to increase the carbon concentration in mineral soils at a rate of $100\times g\,\mathrm{m}^{-2}\,\mathrm{a}^{-1}$ [6]. Moreover, approximately 31% of the land surface of the planet is covered by forests [7], representing approximately 38% in the EU and 36.5% in Spain [8], the country where this study was carried out. Additionally, global forest cover has increased by 7% since the 1980s [9]; this increase is directly caused by the abandonment of agricultural land where it is difficult to mechanize processes to develop modern agriculture [10]. The abandonment of this agricultural land has mainly occurred in districts already dominated by forests [11]. Spain is not an exception, however: reforestation has decreased in the last 10 years, and in 2007, 54,000 ha of forested areas were repopulated, while in 2017, this increase was approximately 12,000 ha [12]. It is important to highlight the economic support that forestry generates in today's society, either as a livelihood or as an economic complement. It is a broad sector that encompasses different activities and industries. European forest-based industries include woodworking, manufacturing pulp, paper and paper products, furniture, printing and bioenergy [13]. Together, these

industries comprise approximately 420,000 companies with a total turnover of more than 520,000 million EUR [14]. For all these reasons stated, sustainable management and proper policies are essential for maintaining the ecological and socioeconomic functions of forest ecosystems [15]. To reach this goal, international forest certification processes and systems (FCSs) have been developed. The total area of certified forests worldwide steadily increased from less than 25 million hectares in 1998 to 405 million hectares in 2018, comprising 11% of the world's forests [16]. In this sense, a requirement for the sustainable development of forest management is to ensure the ecological security of forests [17].

Therefore, multitemporal observations can be useful tools to detect changes in forest ecosystems. In the face of rapid and multiple forest changes, remote sensing has become the most practical and efficient means of extracting information with great temporal, spatial and thematic detail [18]. Consequently, multiple detection methods have been developed to monitor changes in forest areas. Most of these methods involve bitemporal images [19], and the newest method continuously records the dynamics of disturbances in a dense remotely sensed time series [20], making it easier to track rapid forest changes, such as fires. Regardless of the specific process, these remote sensing methods outperform traditional methods [21]. Hyperspectral sensors, which monitor the Earth's surface in contiguous and narrow bands, allow us to capture the biochemical composition of vegetation [22], providing a significant level of detail. However, an optimal set must be selected from the broad set of wave bands, as most of the bands are highly correlated and require large amounts of computational power [23]. Generally, when very-high-spatial-resolution images are used, the spectral responses of individual trees are affected by differences in canopy illumination and background signals [24]. For this reason, in vegetation studies that cover large geographic areas, relatively dense and freely accessible multispectral imagery, such as Sentinel-2, appears to be the best solution [25,26]. Despite the wide availability of data, few studies have been conducted focusing on characteristic smallholding plots in which temporal analyses conducted using Sentinel-2 data to analyze forest vegetation and deal with the detection of cutting in a wide territory [27,28] were the objectives. Studies using satellite imagery have focused on the deforestation or degradation of ecosystems [18,29], especially in large areas such as forests where deforestation affects highly valuable ecosystems, and these images have also been used to analyze vegetation segmentation [30,31]. However, it is necessary to develop a methodology that, in the face of the rapid and multiple changes that occur in forests due to harvesting and forestry activities, analyzes information at the plot level. In addition, this method must contribute to and facilitate the planning and execution of the extraction of wood to control and guarantee good practices and guarantee the objectives of good forest use. In other words, the development of a methodology that is adapted to a given study area will enable forest managers to recognize when cutting has been carried out in that area. In this sense, the mapping of forest changes will provide information on the potential for obtaining wood, patterning forest growth and detecting risks associated with the multiple impacts that these ecosystems may suffer.

Focusing on the regional object of this study, Galicia is the most important forest region in Spain, with 8% of the total Spanish forest area [32]. This region contains 2,040,754 million ha, wherein almost 1,500,000 million ha belongs to wooded areas, with 28% conifers, 51% hardwoods and 21% mixed-use forests [33]. *Eucalyptus globulus* and *Pinus pinaster* are the most widespread forest species in this area due to commercial interest. Forest management and industry account for 12% of the final agricultural production in the rural economy of Galicia, compared to 3.5% in Spain overall. Forestry and wood transformation contribute 13% and 43%, respectively, to the gross domestic product (GDP) of Galicia, and together, these processes generate almost 3% of the total employment of this region [34]. On the other hand, this forested area is characterized by high fragmentation in terms of ownership, and more than two-thirds of the forestlands of Galicia belong to 670,000 owners. Individual forest holdings vary in size between 1.5 and 2.0 ha on average. Private forests comprise 97.3% of Galician forestland, and this land is often divided into up to 10 noncontiguous parcels, yielding a mean surface area of only 0.26 ha [35]. Two-thirds of the land under this

private ownership belongs to individual properties, and the remaining third is collectively owned by residents through Communities of Communal Forests. These dispersed and very small properties are one of the main obstacles preventing the further development of the sector, although great efforts have been made in recent decades to overcome this structural problem through the consolidation of territories and the creation of forest societies [35]. In addition to this high dispersion problem, the low profitability caused by a lack of mechanization caused by the situation of plots on slopes, poor forestry or the volume of felling poses additional issues [36,37]. In addition, the low technical qualifications of owners, together with their advanced age, induce the low innovation and development of the sector [38]. Finally, it is necessary to highlight some of the impacts that these owners suffer the most, such as forest fires [39] and the presence of pests and insect diseases [40].

In this social, economic and environmental context, this work has the objective of verifying the potential of Sentinel-2 time-series images as key information in the process of the detection and temporary control of very-small-plot forest cuts within a management and control system. The priority of this study involves checking the level of effectiveness of these images and the influence that the size, shape, number and casuistry of the composition of the plots have on the detection of forest cuts. Thresholds are established to indicate, with high probability, the existence of felling, as well as the precision of this technology. At the same time, the influence of the irregularity of the applied time series resulting from the absence of information caused by the presence of clouds or other problems is analyzed. This innovative research focuses on the characteristics of the plots, and small parcels are designated to determine whether the proposed methodology is valid for smallholdings. The results achieved through this research could be integrated into the control process of a forest management group made up of thousands of owners and plots.

## 2. Materials and Methods

### 2.1. Study Area

This study was carried out in Galicia (Northwestern Spain) (Figure 1). This region comprises an area of 29,575 km$^2$ and has a population of almost three million people [41]. It has a Mediterranean oceanic climate with mild summers [42]. The predominant soils are developed on granitic rock and acid schist, have a loamy or sandy loam texture and are well-drained.

The selected plots come from the forest management database of the local company Asefor Ingeniería Forestal, S.L. From this database, the plots that are within the limit of Tile 29TNH and are smaller than 3 hectares with clear-cutting, cleaning or forest cutting in general occurring between 2018 and 2020 were selected as the study area. Of these plots, all the information regarding their characteristics and uses is known thanks to previous inventory work carried out by the company. Small parcels were designated to determine whether the proposed methodology is valid for smallholdings. In total, 4231 cadastral parcels (Figure 1), that is, individual and separate plots, were considered. This number of pilot plots was considered sufficient to verify the operability and adaptability of the developed control method. The plots have an average area of 0.47 ha, with a minimum area of 0.0034 ha and a maximum of 3 ha. The average perimeter of these plots is 324 m, with a minimum of 27 m and a maximum of 1545 m. An area of 1,157,000 ha, divided into pixels of $10 \times 10$ m, was analyzed and treated with the results presented in the subsequent sections and assigned to each of the 4231 cadastral parcels. All the plots contained forest species that were cut down at some point. Only species of the genera *Pinus* and *Eucalyptus* were cut. Some plots that were analyzed were not cut in their entire area; this may occur for multiple reasons, the most common of which are described below. Some plots contain several species (Figure 2b); therefore, two-aged stands (growing areas with trees of two distinct age classes [43]) or uneven-aged stands (stands of trees of three or more distinct age classes, either intimately mixed or in groups [43]) were considered. Other plots, despite having the same forest species, had different age classes (Figure 2a). Therefore, these plots had two-aged or uneven-aged silvicultural systems. Finally, some plots, or those that were also

found in the two previous cases, contained "non-cuttable" forest species such as deciduous hardwoods, unique species or riparian vegetation. On the other hand, some of these plots were not entirely forested, sharing their areas with agricultural uses (meadows and crops), a condition that may affect the detection of cuts due to changes in vegetation that are typical of crop rotation.

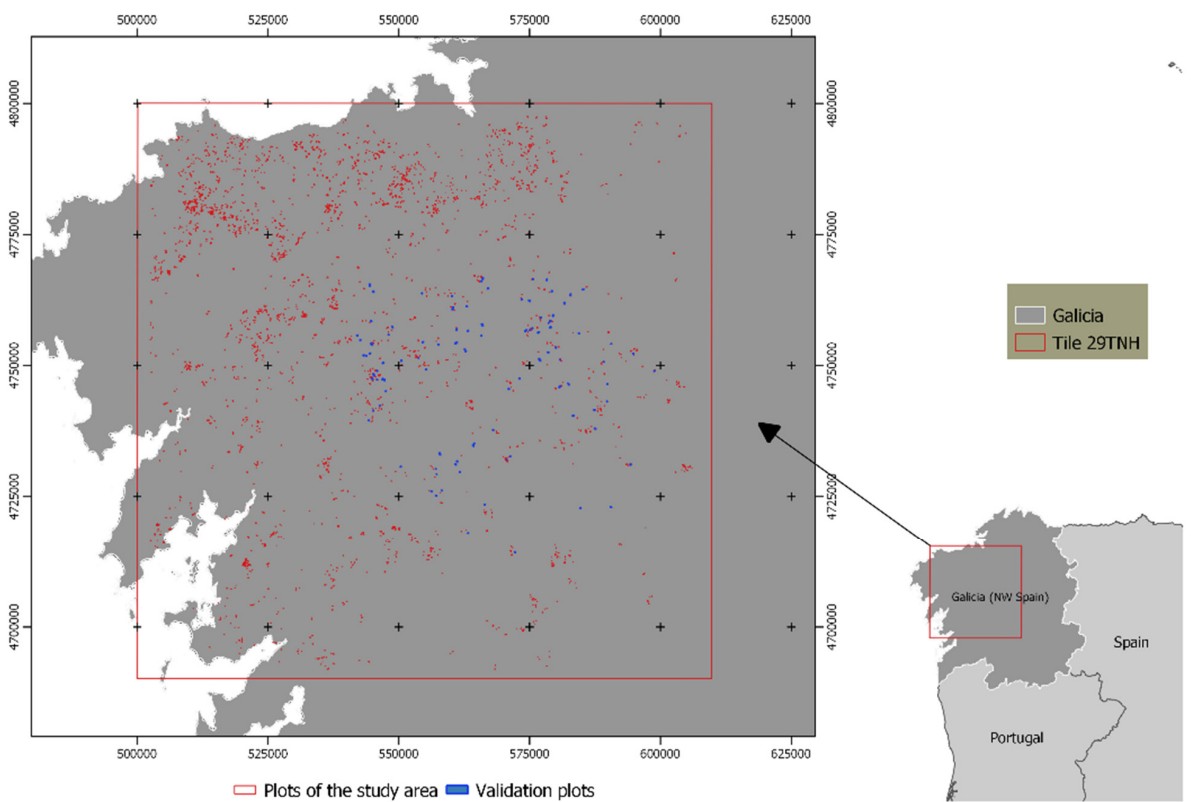

**Figure 1.** Analyzed plot distribution in Galicia (NW Spain). Coordinate system ETRS89-UTM29T.

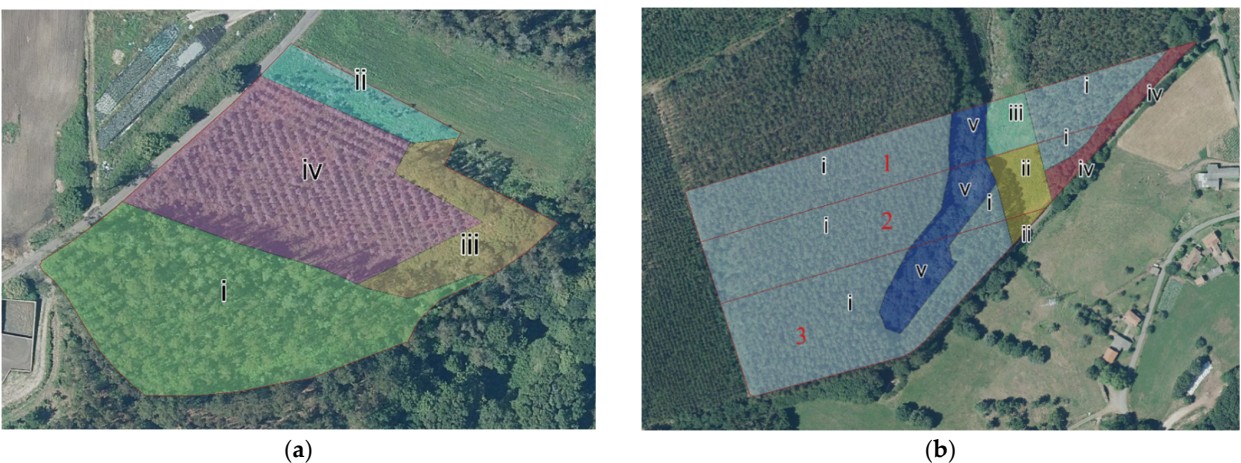

**Figure 2.** Example of some plots with different uses, forest species and cutting ages. (**a**) Plot of 1.26 ha and (i) reforestation of *E. globulus* close to the cutting age (0.56 ha); (ii) recent reforestation of *E. globulus* less than 1 year old (0.08 ha); (iii) scattered deciduous hardwoods with scrub (0.15); (iv) repopulation of 5-year-old *E. globulus* (0.47). (**b**) Three plots with (i) reforestation of *E. globulus*, regular masses of the same year (plot 1: 1.5 ha; plot 2: 1.36 ha; plot 3: 1.68 ha); (ii) pasture forage crop (plot 2: 0.26 ha; plot 3: 0.04 ha); (iii) scrub (plot 1: 0.2 ha); (ii) and (iii) tree covers cleared by a power line); (iv) *Quercus robur* (plot 1: 0.12 ha; plot 2: 0.12 ha); (v) riparian vegetation (plot 1: 0.16 ha; plot 2: 0.25 ha; plot 3: 0.3 ha).



*2.2. Data Collection and Preprocessing*

The dataset used in this study came from a selection of images obtained from the standard Sentinel-2 Level-2A. The Sentinel satellite carries an innovative wide-swath, high-resolution, multispectral instrument (MSI) with 13 spectral bands with wavelengths ranging from 440 to 2200 nm and from 10 to 60 m depending on the spectral band [44]; in this case, the resolution used was 10 m. The Sentinel-2 Level-2A product comprises 100 km × 100 km mosaics in the UTM/WGS84 projection [45] and provides orthorectified images with reflectance levels below the atmosphere (BOA) [46].

Sentinel-2 imagery collected between January 2018 and December 2020 was downloaded from the Copernicus Open Access Hub [47]. Initially, images with cloud percentages equal to or less than 5% in the selected time period were searched. On dates for which this condition was not met, due to the absence of images with less than 5% cloud cover, the condition was increased to a maximum of 15% cloudiness. The images were processed with the free software QGIS 3.8.2 using the SCP (semiautomatic classification) tool. An orthophoto was obtained from the Spanish National Geographic Institute's PNOA (National Aerial Orthophotography Plan) [48]. Subsequently, each of the images was visually analyzed, and the images that exhibited cloudiness in the plots under analysis were discarded, as were those that were not within the coverage of the Tile 29TNH. The presence of elements in the images that hinder the correct analysis of data, such as aerosols, cloud shadows or the presence of fog in river valleys, must be analyzed and solved to guarantee the correct control of forest cutting. In total, 54 images were acquired for the selected period (Table 1). The data were visualized in QGIS and geospatially managed in the PostGIS database manager.

**Table 1.** Images acquired from the Copernicus Open Access Hub for the time period analyzed from 2018 to 2019.

| 2018 | 2019 | 2020 |
|---|---|---|
| 24/02/2018 | 14/02/2019 | 24/02/2020 |
| No data | 16/03/2019 | 18/03/2020 25/03/2020 |
| 18/04/2018 | No data | 14/04/2020 24/04/2020 |
| 05/05/2018 18/05/2018 | 13/05/2019 | 19/05/2020 24/05/2020 27/05/2020 29/05/2020 |
| 17/06/2018 19/06/2018 24/06/2018 | No data | No data |
| 09/07/2018 | 12/07/2019 | No data |
| 01/08/2018 21/08/2018 26/08/2018 | 21/08/2019 | 05/08/2020 |
| 07/9/2018 | 15/09/2019 | 29/09/2020 |
| 10/09/2018 10/10/2018 20/10/2018 22/10/2018 | 07/10/2019 10/10/2019 22/10/2019 | 16/10/2020 |
| 14/11/2018 | No data | 13/11/2020 20/11/2020 30/11/2020 |
| 26/12/2018 | 04/12/2019 26/12/2019 | No data |

After the selection of images that were as cloud-free as possible, preprocessing was carried out to eliminate the presence of clouds or cloud shadows that were in the range of the studied plots. Pixels with blue band reflectances below 0.01 were considered to be predominantly related to undetected cloud shadows [49]. For this reason, the blue band (B2) of each image was used to eliminate pixels above the 1150 spectral profile. The objective of this process was that for each date analyzed, a cloud mask could be created that was omitted from the processing; in this way, atmospheric pollution that affected the calculation of the NDVI index was eliminated. Although this process is not perfect, as it does not totally eliminate all types of cloudiness or fog, it was valid for eliminating a large portion of the subsequent anomalies. In this study, only forested plots were worked on, and the elimination of other land use types (urban, water, etc.) did not affect the object of this methodology. The threshold of 1150 was obtained by performing systematic sampling in places where clouds were detected, and this value was chosen after several visual tests of the obtained results were performed.

### 2.3. Normalized Difference Vegetation Index (NDVI)

The spectral index used in this study was the normalized difference vegetation index (NDVI) [50]. If NDVI is compared with other vegetation indices, it has the best correlation with canopy cover, especially in the dry season [51]. NDVI is the most reliable vegetation index used to estimate forest density [52,53]. For these reasons, it was selected for use in this study. The description of NDVI is as follows:

$$\frac{\rho_{NIR} - \rho_{Red}}{\rho_{NIR} + \rho_{Red}} = \frac{B8 - B4}{B8 + B4} \tag{1}$$

where *NIR* refers to the spectral reflectance at near-infrared wavelengths (0.85–0.88 μm), and RED refers to the spectral reflectance at red wavelengths (0.64–0.67 μm) .

The aim of applying this index was to study how NDVI variations occur in different forest plots and to analyze the capacity of the proposed method to detect abrupt changes in forest cover, such as clear-cuts, cleaning and forest cuts in general. The mean NDVI value was obtained for each plot in each image through RStudio [54]; the same software was used for the generation and presentation of raster images. This process allows the spatial and temporal characteristics of these changes to be detected. Through this analysis with RStudio, each plot was temporally studied on each date (Table 1), and the phenology present in each plot was statistically studied, as were the changes caused by cuts. In addition, the processes were automated through the use of scripts with RStudio. The 4231 felling plots were reforested by the *Pinus* and *Eucalyptus* genera, although in many of the plots, other species or intercropped covers (mainly deciduous leafy plants, scrub, pastures, etc.) existed that were not cut, as was mentioned already; these plots experienced differing evolution over the course of the annuity and seasonality. *Pinus* and *Eucalyptus* have no marked seasonality, so the differences will be taken as periods. Therefore, abrupt changes in NDVI in one pixel are expected to be due to cutting in most cases. NDVI can help identify and quantify changes in the analyzed forest plots. To accomplish this, the differences in NDVI between different periods were calculated using the following equation:

$$NDVI(t) - NDVI(t - k) \tag{2}$$

where "*NDVI*" is the value of the raster at each point, "*t*" is the initial period of analysis and "*k*" is the number of previous periods with which the most recent image is compared. Generally, in this case study, *k* is equal to 1, as the most recent image is always compared with the immediately previous period. An exception occurred when, in cases where no image was available for the previous period, it was necessary to compare the most recent image with images from previous dates.

All the locations of the selected pixels and their values were stored in tables managed in PostGIS for later treatment in RStudio and visualization in QGIS.

### 2.4. Detection of the Forest Cutting Threshold

To detect the values that allow the probability of felling to be discerned, a statistical analysis was carried out. The packages used to carry out the classification tree were: library *rpart*, *rpart plot* and *caret*. In this process, NDVI differences were established for each satellite information period. All trend data were extracted in 53 time periods with 197,398 pixels analyzed. A statistical boxplot analysis was performed for all pixel trends for each plot over the period analyzed in this study (Figure 3). The lower extreme was then used to identify the cut plots. This extreme represents the global minimum value used for the detection. The value used was −0.1254; that is, a pixel was "cut" when the trend drop between periods was ≤ −0.1254. The detected threshold values for the trend drops for which cuts could be detected were obtained for the set of all pixels of all the plots regardless of the coverage of the individual plots. This analysis also provided a median value (0.0009), lower extreme value (−0.1254) and highest (0.1279) and lowest extreme values (−0.1254).

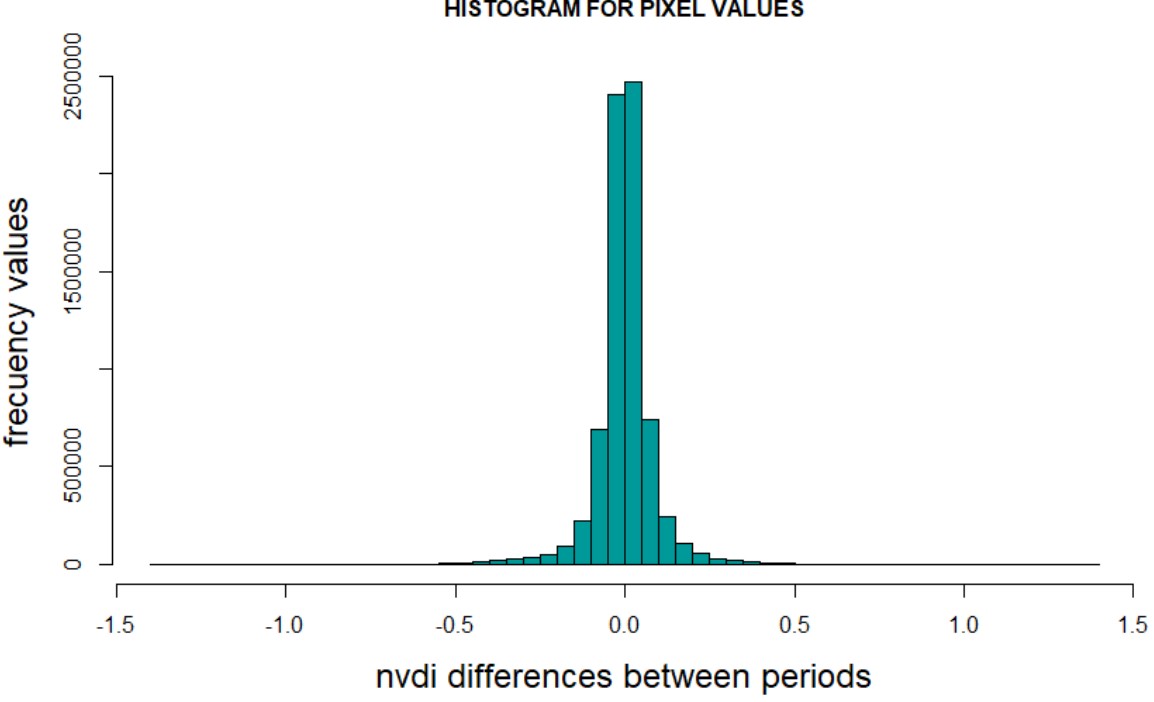

**Figure 3.** Frequency distribution of the 197,398 values with their trends for the 53 periods analyzed.

The locations of the pixels that met the threshold conditions defined in each period were related to the surfaces. A database was obtained, including the period, identification of the plot and fall surface and its percentage in relation to the surface of each plot. This information was compared with RGB images. A total of 135 plots were selected and statistically analyzed). When felling is detected, statistical analyses are carried out for pixels below −0.12: mean of the values, twenty-fifth percentile, fiftieth percentile, seventy-fifth percentile, the area with variation and the relationship of this area with that of the total area of the plot. The same values were also studied for the same plot in the periods in which no short is detected. These analyses (Appendix A) are used to create the database that will form the classification tree. A decision tree or classification tree is a supervised matching-learning algorithm that will show us a series of sequential decisions [55]. This classification tree was constructed (Figure 4) with all the data collected as a prediction model by obtaining an algorithm whose objective (dependent variable) was the detection of forest cutting; specifically, 1 = "Cutting" or 2 = "No cutting". The aim of the decision tree is to divide a complex decision into several simple decisions. Using this approach,

the statistics defined from the data are predictor variables, while the short one would be the target variable. Since the class proportions of a mixed pixel are measured on a continuous scale from 0 to 1, it is necessary to apply decision tree regression to estimate them. The data set was split in two, with 70% being training data and the remaining 30% being test data, to check the effectiveness of the model. From this 70% training data, an algorithm was obtained, which allowed us to predict the value of the target variable using the independent variables. The *rpart* function is used to train the model, and the target variable, in this case, "*clearcut*", is formulated, while the rest of the variables are used as predictors. The algorithm finds the independent variable that best separated the data into groups; this separation corresponds to a rule, and each rule has a node. The data were then separated (partitioned) into groups based on the obtained rules. Then, for each of the resulting groups, the same process was repeated. It first checks the percentile value, then the ratio between the occupied pixels and finally, the mean pixel values. This process was carried out repeatedly until it was no longer possible to obtain a better separation. When this occurred, the algorithm ended. The model is then tested with the remaining 30% of data, from which the confusion matrix is obtained (Table 2). Then, the data contained in the RGB images were visually reviewed to verify and validate the detection of cuts based on the applied methodology (Figure 5).

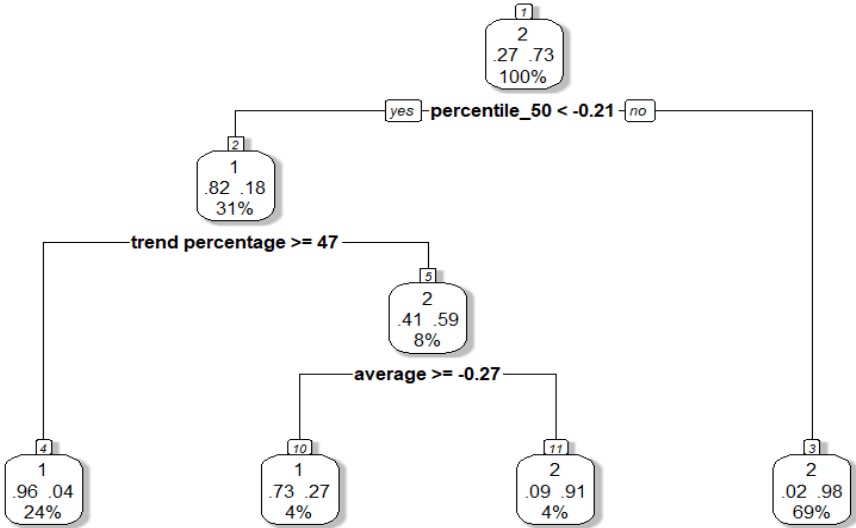

**Figure 4.** Classification tree obtained from the plots analyzed in this study. Each of the rectangles represents a node with its classification rule. Inside the rectangle of each node it is the proportion of cases belong to each category (%) and the proportion of the total data that has been grouped there (%).

**Table 2.** Statistical analysis including equation and description of each indicator.

| Accuracy | $\frac{TP+TN}{TP+FN+TN+FP}$ | $TP$ = true positive, $TN$ = true negative, $FP$ = false positive, $FN$ = false negative |
|---|---|---|
| 95% CI | $\bar{x} \pm 1.96 \frac{Sx}{\sqrt{n}}$ | $\bar{x}$ = the sample mean<br>$Sx$ = margin of error<br>$N$ = standard error |
| *p*-value (ACC < *NIR*) | $z = \frac{\hat{p} - p_0}{\sqrt{\frac{p_0(1-p_0)}{n}}}$ | $\hat{p}$ = the sample proportion<br>$p_0$ = the hypothesized proportion<br>$n$ = the sample size<br>ACC = accuracy<br>NIR = No information rate<br>*p*-value for ACC > *NIR* |

**Table 2.** *Cont.*

| | | |
|---|---|---|
| Kappa | $\frac{p_0 - p_e}{1 - p_e}$ | $P_0$ = proportion of trials in which judges agree $P_e$ = proportion of trials in which agreement would be expected due to chance |
| Mcnemar's Test | $x^2 = \frac{(b-c)^2}{b+c}$ | - |
| Sensitivity | $\frac{TN}{TP+FN}$ | $TP$ = true positive, FN = false negative |
| Specificity | $\frac{TN}{TN+FP}$ | $TN$ = true negative, $FP$ = false positive |
| Pos. Pred Value | $\frac{TP}{TP+FP} \times 100$ | $TP$ = true positive, $FP$ = false positive |
| Neg. Pred Value | $\frac{TN}{TN+FN} \times 100$ | $TN$ = true negative, FN = false negative |
| Prevalence | $I \times D$ | $I$ = incidence, $D$ = duration |
| Detection Rete | $\frac{TP}{TP+FN}$ | $TP$ = true positive, $FP$ = false positive |
| Detection Prevalence | $\frac{T_{detection}}{Total} \times 100$ | - |
| Balanced Accuracy | $\frac{Sensitivity + Specificity}{2}$ | - |

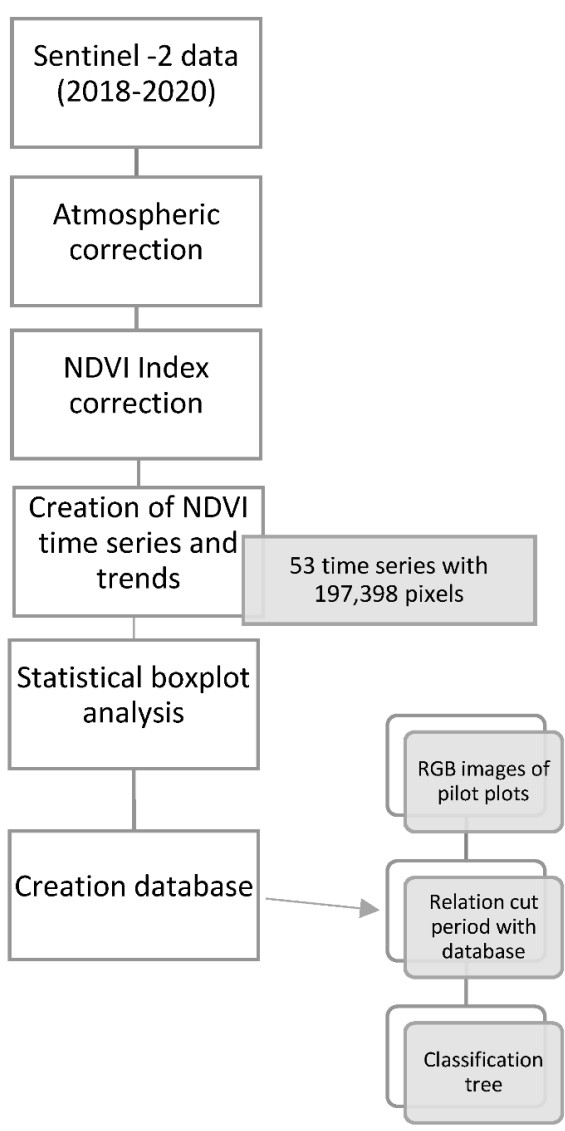

**Figure 5.** Summarizing diagram of the methodology carried out.

## 3. Results

### 3.1. Variation in NDVI

NDVI was calculated for each forest parcel (the two examples in Figures 6 and 7 reflect the variability in the characteristics of the plots). The aim was to analyze the sudden changes observed through these values and to study whether they allow the automatic detection of spatially and temporally limited cuts. Therefore, of the 4231 total plots, there were 197,398 pixels in which NDVI was obtained in 53 different time periods.

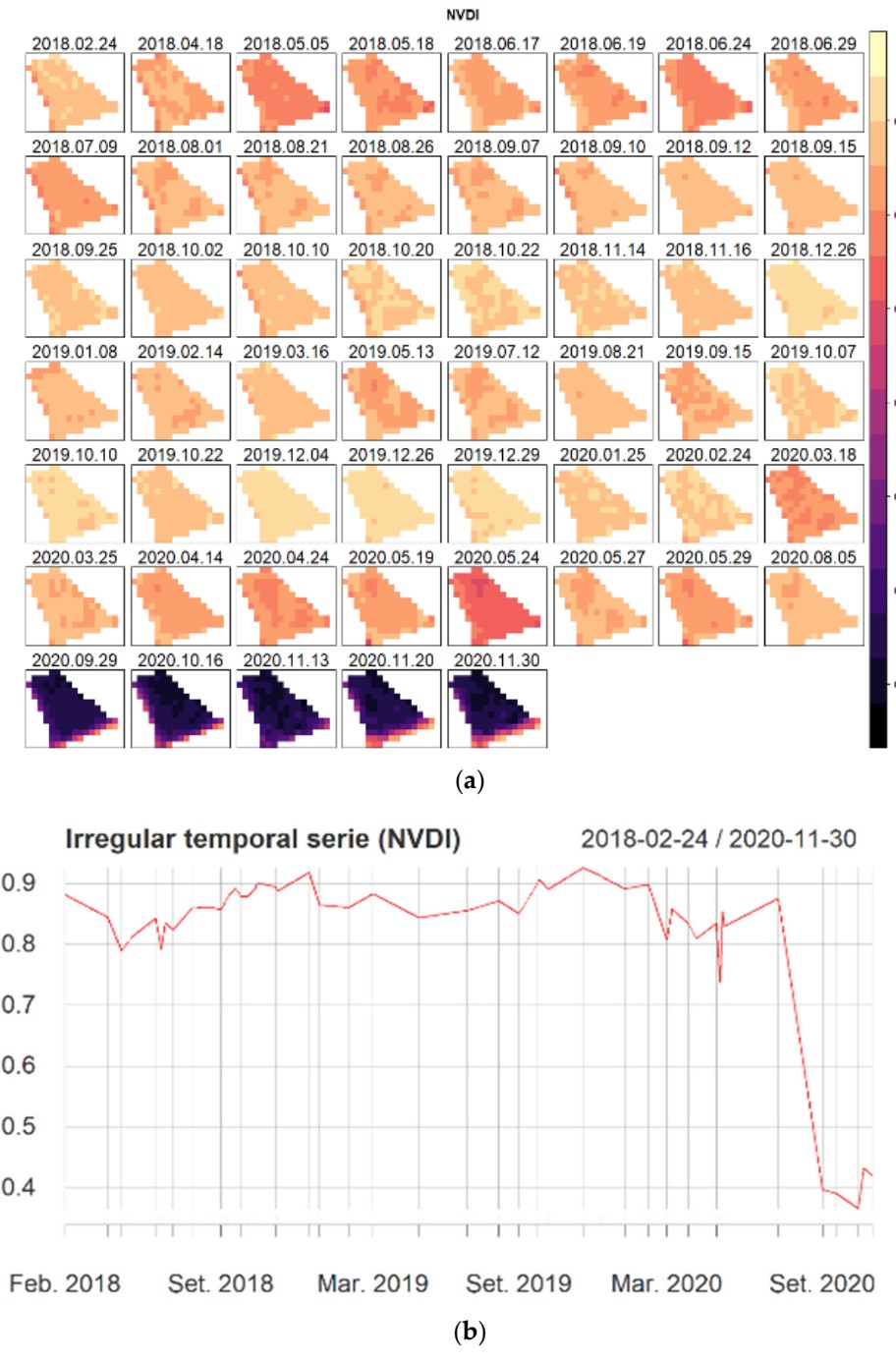

(a)

(b)

**Figure 6.** *Cont.*

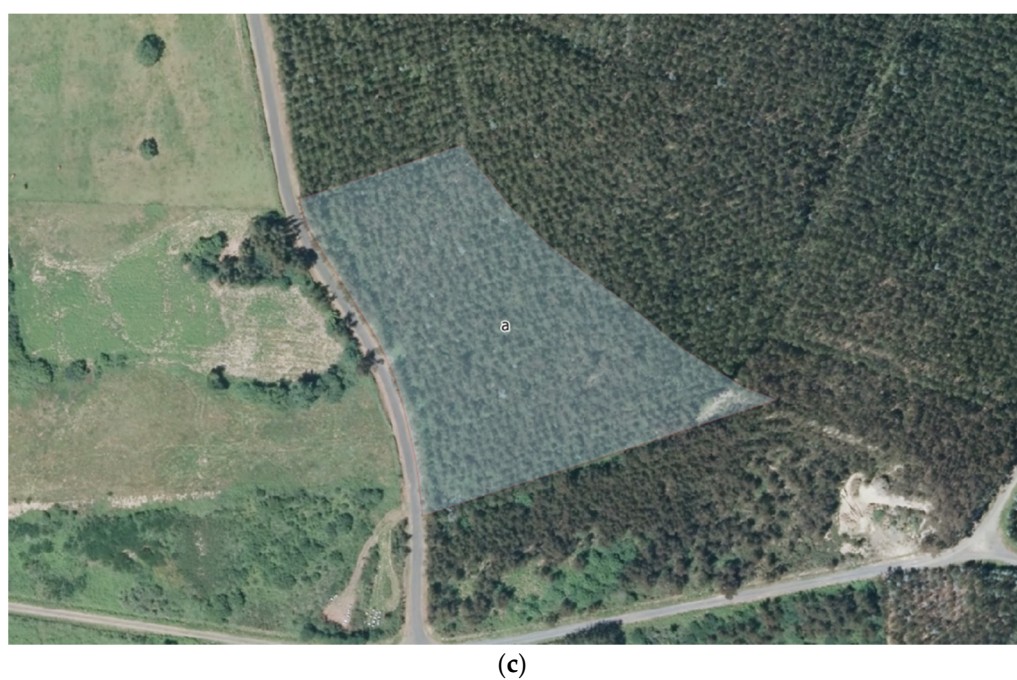

(**c**)

**Figure 6.** (**a**,**b**) Spatial and temporal variations in NDVI in a pilot plot of *E. globulus* with an area of 1.08 ha. (**c**) Land uses: *E. globulus*.

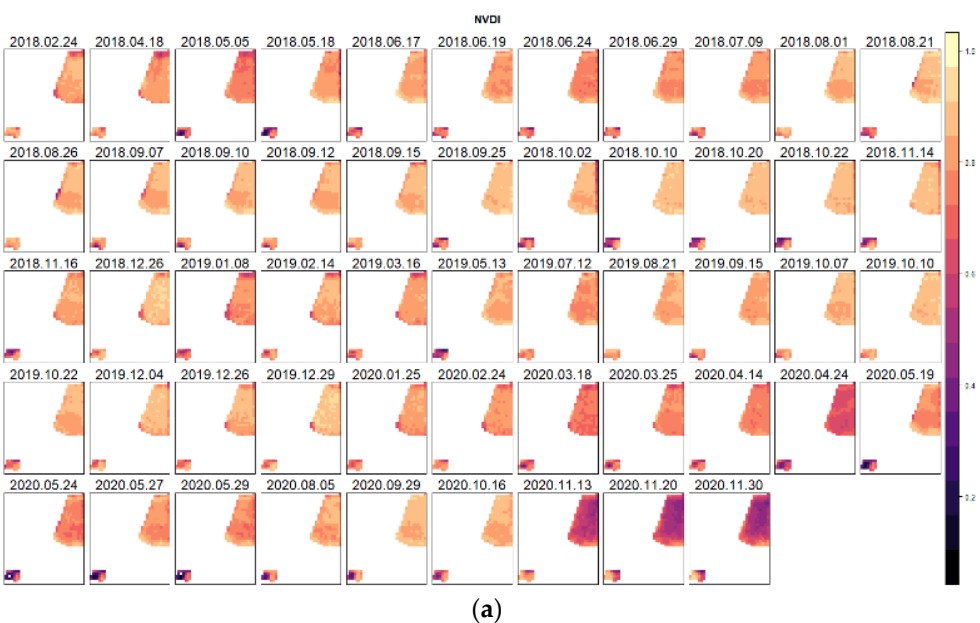

(**a**)

**Figure 7.** *Cont*.

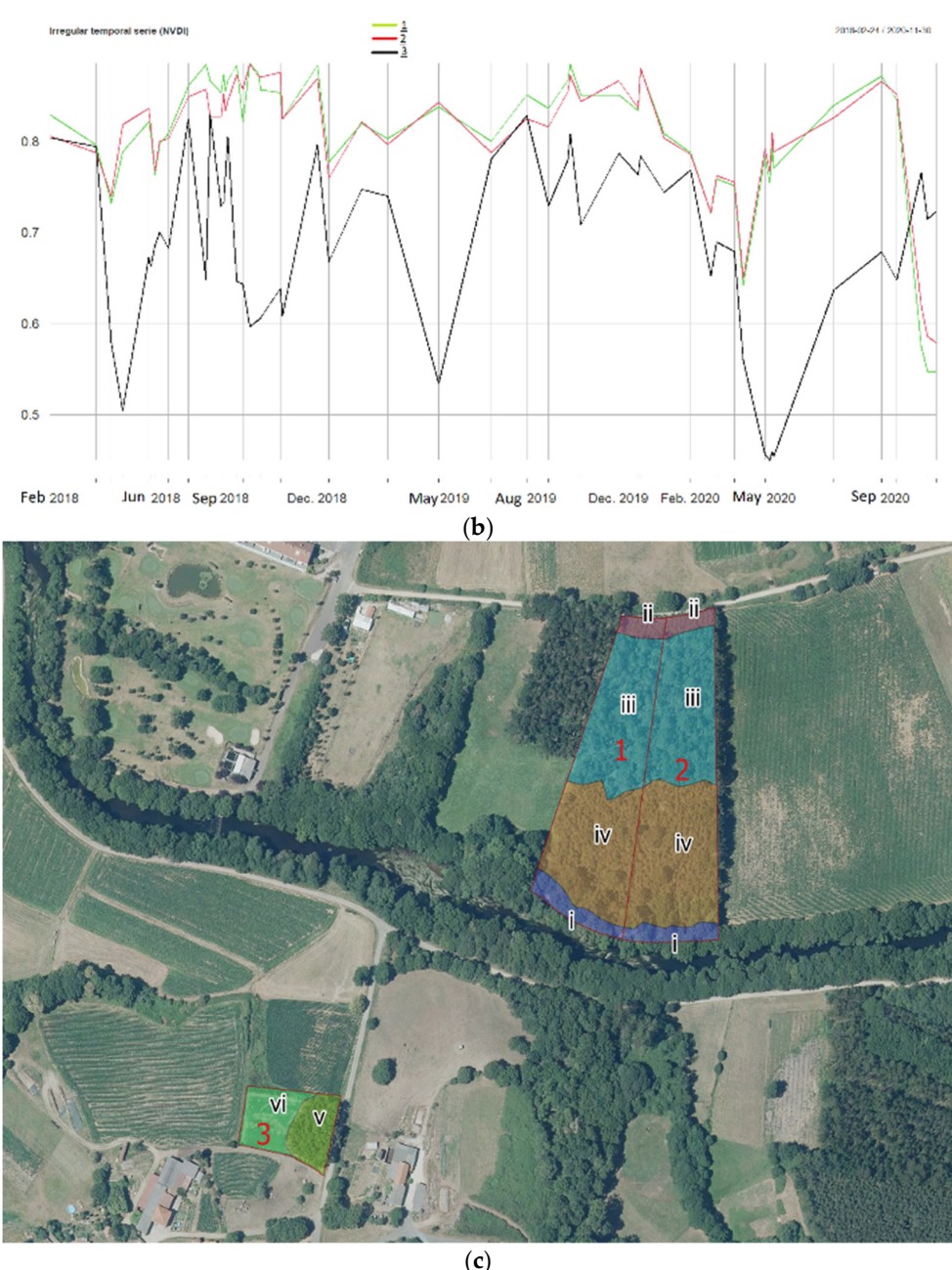

**(b)**

**(c)**

**Figure 7.** (**a**,**b**) Temporal variations in NDVI in three plots with different species and states. (**c**) Land uses in plot N°1: (i) Leafy riparian deciduous (0.07 ha). (ii) Leafy deciduous dominated by *Q. robur* (0.04). (iii) Irregular mass formed by *P. pinaster* and loose stands of *E. globulus* (0.44 ha). (iv) Coniferous plantation with loose, isolated stands of *E. globulus* (0.48 ha). Land uses in plot N°2: (i) Deciduous broadleaved riparian hardwoods (0.08 ha). (ii) Coniferous plantation with loose and isolated stands of *E. globulus* (0.48 ha). (iii) Irregular mass consisting of *P. pinaster* and single stands of *E. globulus* (0.46 ha). (iv) Deciduous hardwoods dominated by *Q. robur* (0.04 ha). Land uses in plot N°3: (v) *P. radiata* plantation (0.13 ha). (vi) Grassland and cultivation (0.16 ha).

The temporal information obtained for each plot allowed the correct overall analysis of the biomass of each plot. The different felled plots were mainly reforested by *Pinus* and *Eucalyptus*, although in many plots, other species (mainly deciduous hardwoods, scrub, grasses, etc.) were mixed that were not felled and that had a different evolution over the course of the year and a different seasonality. In addition to displaying the felling characteristics of plots, obtaining the temporal analysis made it possible to reveal the phenological changes of the species in each plot. This phenomenon can be clearly observed in Figure 5a,b, where along the temporal analysis, slight fluctuations were caused by phenological changes. Finally, the last images of the time series show the felling of the plots. Plot N°2 shows a partial cutting in October 2019 and the recovery of the vegetation in the following months.

### 3.2. Detection of the Forest Cutting Threshold

From all the images analyzed, 53 time periods were analyzed for each plot, with a total of 197,398 analyzed pixels. With the aim of explaining the analysis carried out, one pilot plot was selected to show the detection capacity of the applied methodology (Figure 8a). A histogram representing the frequency of each of the pixel values along the analyzed time series (Figure 8b) confirmed the previously selected cut detection value (−0.1254). In addition, with the boxplot analysis (Figure 8c) of each pixel found within the plot, the variations in their values along the time series allowed the detection of the time at which the felling took place.

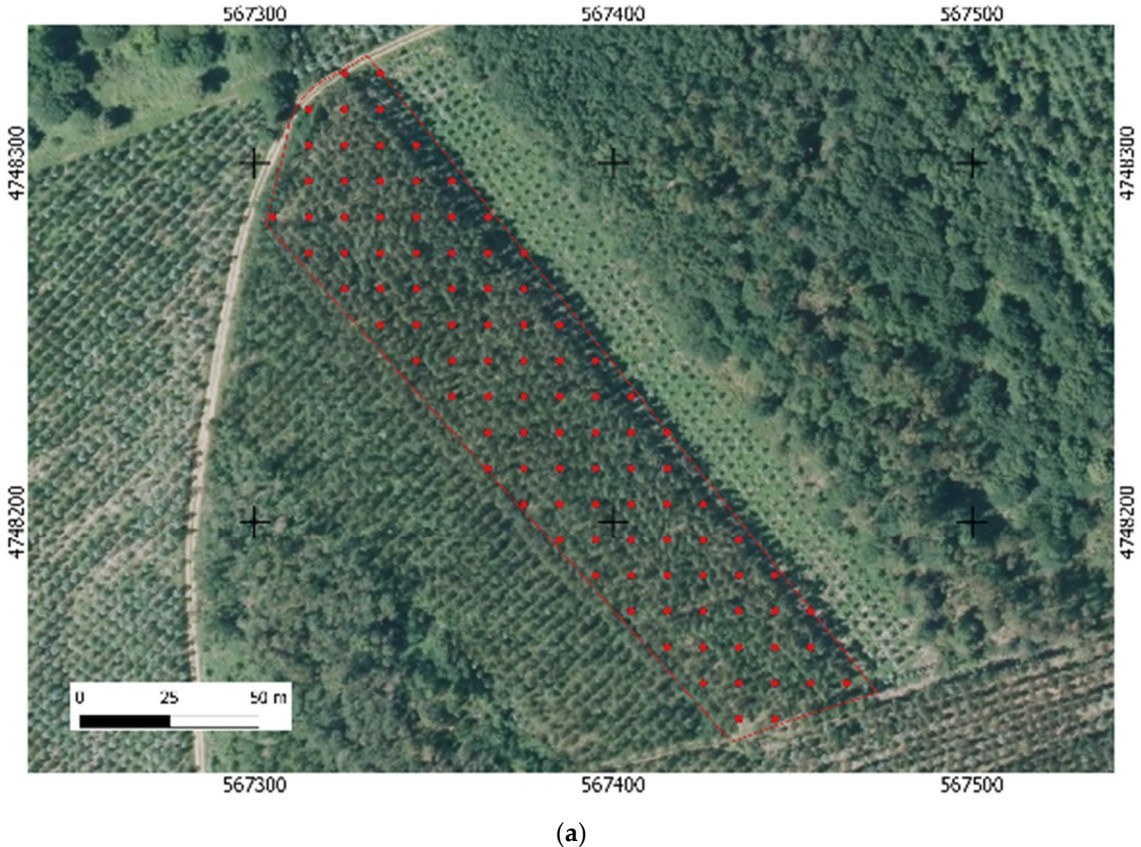

(**a**)

**Figure 8.** *Cont.*

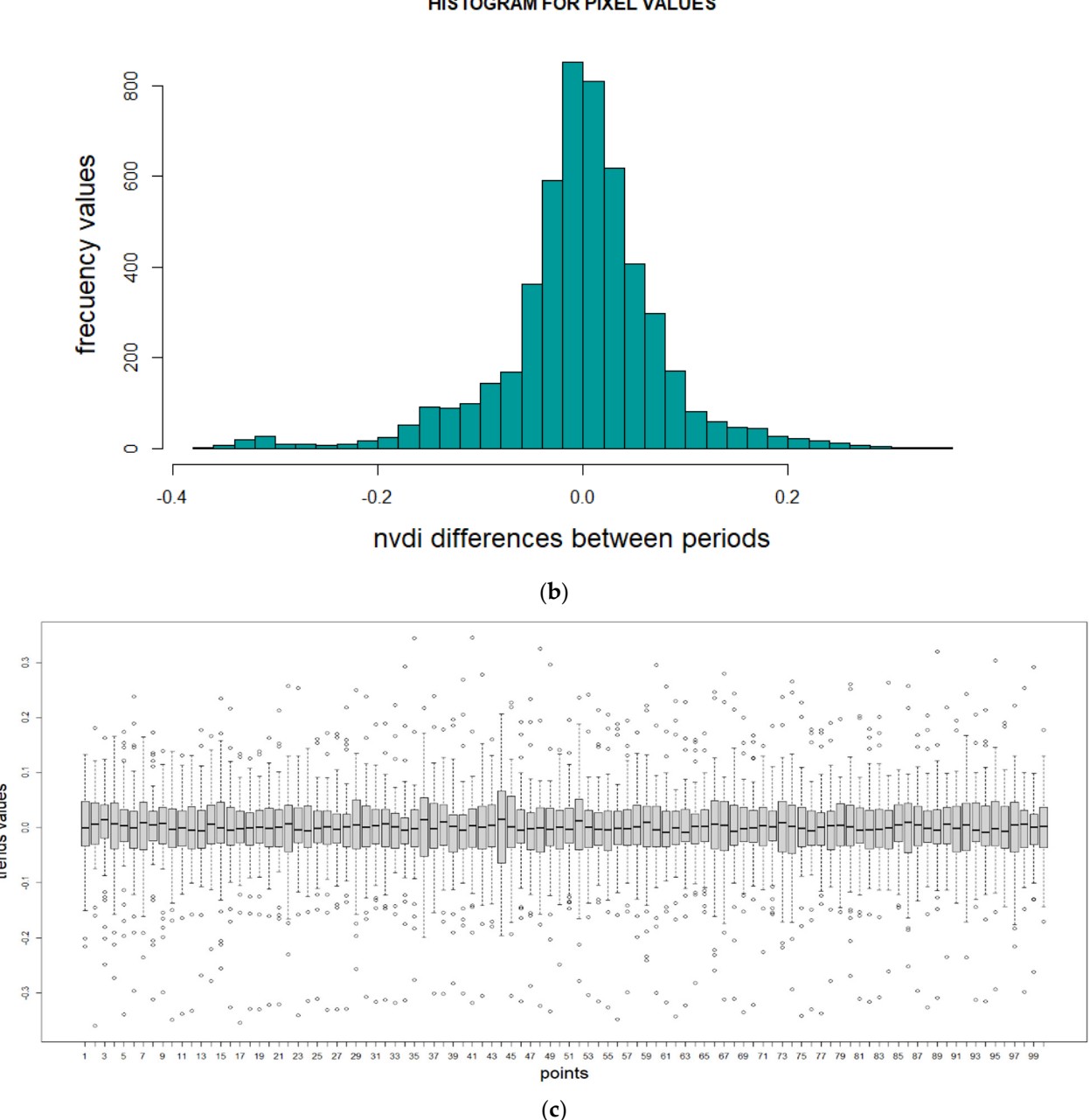

**Figure 8.** Example plot of *Eucalyptus* with an area of 0.99 ha. (**a**) Location of the center pixel (100) where the information of the analyzed period (53) was stored. (**b**) Histogram representing the frequency of each of the pixel values along the analyzed time series. (**c**) Boxplot analysis of each pixel found within the plot and variations in their values along the time series.

From the 135 selected plots, the decreases in NDVI in each period due to a cut or a "not cut" (caused by phenological changes) were detected. Moreover, if sufficient contiguous images were available, it was possible to estimate the duration of forest action. Estimates of the cut area could even be made daily, as could estimates of the total final area of the felling action. In this example (Figure 9), four different plots were analyzed, and a total felling area of 4.42 ha was estimated to have occurred between the end of December 2019 and February 2020. In this example, the cut began on 16 July 2019 (Figure 9a), with 58 points showing felling due to differences in NDVI values between two periods, denoting the detection of a cut with an estimated surface of 0.57 ha, comprising the entire plot. The forest action continued on 4 December 2019 (Figure 9b), affecting 0.5 ha (92.6% with 50 points with decreased differences in NDVI). Another cut was detected on 26 December 2020 (Figure 9c),

with a cut of almost 52% over the total surface area (154 points and 1.54 ha), ending on 24 February 2020 (Figure 9d), with 61% of the area cut (181 points and 1.81 ha).

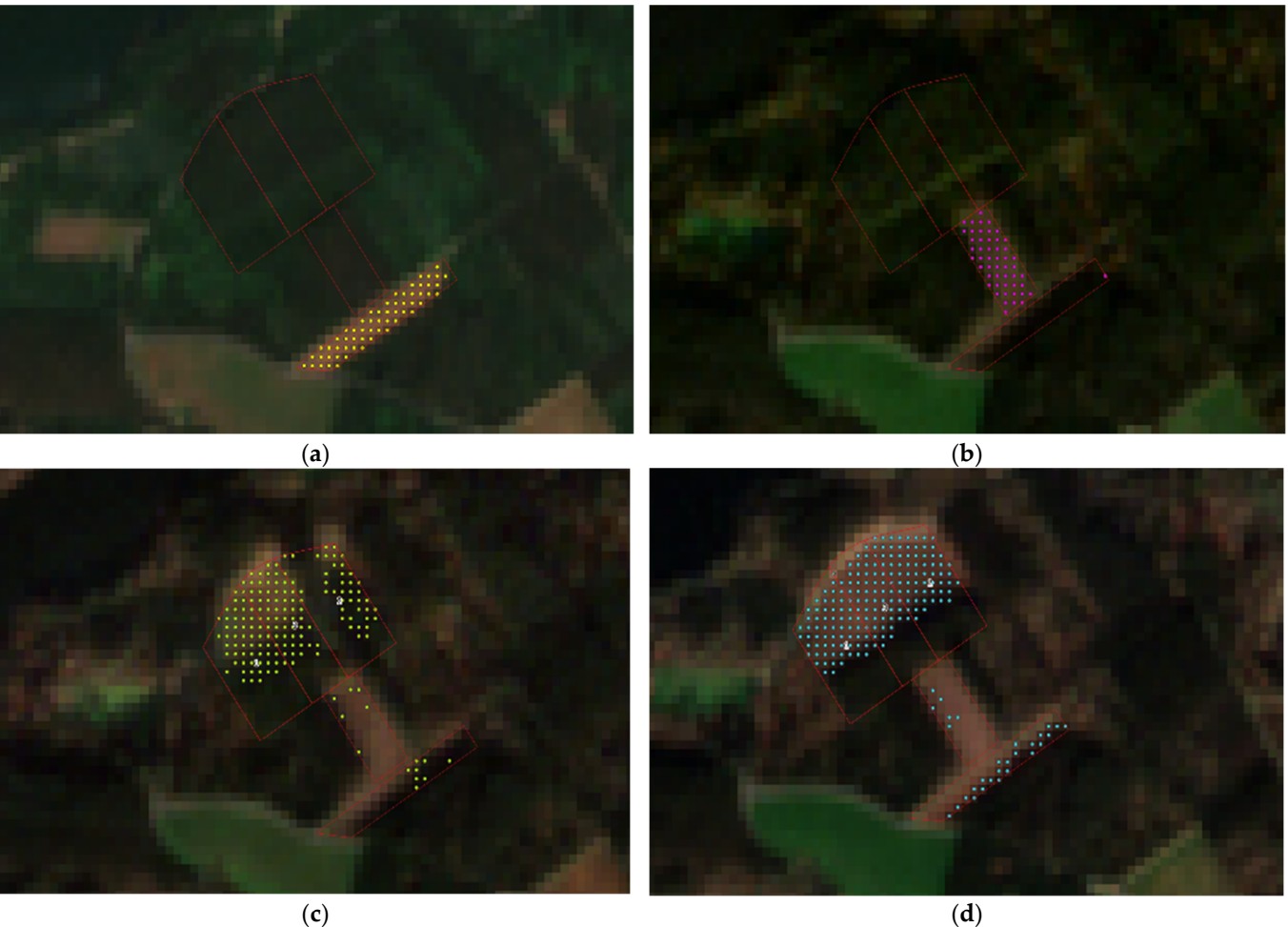

**Figure 9.** Cut detection in four plots of *E. globulus* with different ages and felling shifts. Current high-resolution RGB image with the location and dates of the pixel drops that were identified as cuts during the analysis period. Cutting start date: (**a**) 2019-07-12; (**b**) 2019-12-04; (**c**) 2020-01-25; (**d**) 2020-02-24.

The statistical values of the falls (trends) of the plots were considered and correlated for each period in the selected pilot plot (Table 3). In total, cutting was detected on 111 plots, and 150 falls from periods identified as "not cut" were also detected. In the obtained classification tree (Figure 4), each rectangle represents a node with a classification rule. The rectangle of each node shows the proportion of cases belonging to each category and the proportion of the total data that are grouped together. For example, the rectangle at the bottom left of the graph shows 96% of cases in Type 1 (Cutting) and 4% in Type 2 (Not cutting), representing 24% of all data. These proportions provide the accuracy of the model when making predictions. Thus, the rules leading to the rectangle described above provide 88% correct classifications (Table 3). The classification tree indicates that the value of the percentile decreases and that the values of the means are important as detection variables. This corroborates the hypothesis that the trend percentage (relation between the surface of the felling points and the surface of the plot) can be used for the detection of cuts. The confusion matrix indicates that the model correctly predicts 25 of the 34 total cuts, while 9 are classified as "not cut". On the other hand, the model correctly predicts 84 cuts, while 5 are predicted as cut but are not.

**Table 3.** Confusion matrix used for the classification of forest cutting on the analyzed plots in the study area.

| Confusion Matrix and Statistics | |
| --- | --- |
| Accuracy | 0.8862 |
| 95% CI | (0.8164, 0.9364) |
| No Information Rate | 0.7236 |
| *p*-value (ACC < *NIR*) | $1.044 \times 10^{-5}$ |
| Kappa | 0.7047 |
| Mcnemar's Test | 0.4227 |
| Sensitivity | 0.7353 |
| Specificity | 0.9438 |
| Pos. Pred Value | 0.8333 |
| Neg. Pred Value | 0.9032 |
| Prevalence | 0.2764 |
| Detection Rete | 0.2033 |
| Detection Prevalence | 0.2439 |
| Balanced Accuracy | 0.8396 |

## 4. Discussion

One of the main challenges faced when mapping land cover using time series images is the lack of continuity caused by cloud cover. Sentinel-2 is vulnerable to cloudy weather, making it difficult to obtain sufficient clear images for monitoring in temperate climate zones. However, as pointed out by Vuolo et al. [56], the widespread use of NDVI in remote sensing as a tool for monitoring forest areas has stimulated the development of models that reduce the noise caused by clouds. Puletti and Bascietto [57] conclude that due to the high resolution of the data provided by Sentinel-2, spectral variability over short time periods (5 days) could be considered negligible. Moreover, this information would be more accurate than that provided by Landsat, whose resolution is lower (30 m) and temporal resolution is higher (16 days) [58]. Taking into account that this study was carried out on a type of property characteristic of smallholding, the lack of more frequent images due to cloudiness results in a small decrease in the value of the technology. On such small plots, forest felling can be completed in a single day; the felling of contiguous plots can be completed in several days (Figure 6). Due to the cloudiness problem, the first cut could be detected days later, or it could be assumed that the cut was made in the same time range when more than one plot is cut. In this case, to reduce the noise caused by cloudiness, additional masking was performed in this study using the threshold values obtained for Band 2, as tested in other studies for the additional removal of dense clouds [59]. In this way, interference and cloud contamination were reduced, although it should be highlighted that in a real control system, it is not always possible to eliminate cloud contamination in images. Considering this possibility, this study analyses a long period of time in the absence of images, allowing us to evaluate the behavior of the method in a real control period. In any case, it was verified that a drop in NDVI was detected despite the existence of very small plots or the image being taken days after cutting, implying an acceptable temporal detection margin for sustainable forest management.

This result would also imply the need for high-capacity data storage and analysis. Therefore, whether it is necessary to create large databases and use processing systems that cover these bases must be resolved, taking into account the objective that is being pursued. The use of multitemporal data may have inadvertently increased the noise level in the classification process, as increasing the dimensionality of the data leads to greater redundancy [60]. It would not be implausible that only the use of Sentinel-2 scenes with

optimal timing and atmospheric correction provides good results [61]. However, the use of these images would depend on the objectives of the study. In this case, the multitemporal factor would be the most effective way to address the monitoring of such small plots.

On the one hand, the use of multitemporal data allows phenological information to be obtained with greater precision and allows more effective monitoring in which invalid images are discarded [62]. Furthermore, the use of these images on managed land would allow anthropogenic changes to be detected more effectively [63]. On the other hand, the applicability and use of vegetation indices to detect changes in land use must be considered. These indices are very effective in large areas where changes in land use are analyzed, such as deforestation in the Amazon [64]. However, in this study, the objective was to analyze the use of these technologies in sustainable forest management in a large number of small plots (with different stands or different felling periods), specifically for the detection of the logging or cutting of forest stands. In the latter case, NDVI products have shown good results in multiple studies for different types of forest stands [65,66]. Although it has not been the objective of this work, the described method could also be used for the detection of changes in land use as well as for the detection of forest fires, issues that can be raised as possible future lines of research. In this work, since these are parcels that are part of a certification system, changes in land use are not possible, and forest fires have not interfered in the study since no parcel registered this type of event.

In the present study, this index provided data that were consistent with the forest stands of each plot; in all systems, the index showed very similar patterns within the forested areas (Figure 4). The multitemporal NDVI evolution describes the intra-annual patterns in temperate forest climates [67], showing slight intra-annual and interannual variations, consistent with other studies. At the start of the spring season around Apri–May, NDVI increases until it reaches a maximum in July–August, even extending until October. The use of images from the four seasons facilitates the distinction of vegetation classes with different phenological cycles (for example, conifers and deciduous trees) [68]. These interannual variations were characterized by a rapid increase during the initial growth of each stand, from a minimum postharvest value (a few weeks after harvest) to a first maximum value that occurred during the first or second year after planting [69]. Figure 4b shows the results of a partial cutting that occurred around October 2019, decreasing the NDVI value, which recovered in the following months at different speeds depending on the vegetation present in the plots. This rapidly increasing NDVI index in plots where cutting occurred could be due to the presence of shrubs, which would explain why the values remained lower at the end than at the beginning of the study, reaching the highest point in spring 2020. On the other hand, Figure 4—N°1 shows the NDVI index for a Eucalyptus monoculture plot. This tree is evergreen, so it is normal that NDVI remained constant in this stand, with small phenological variations, until the time of cutting. Hua Lu et al. [70] found that the average NDVI values varied from 0.6 to 0.8 for *Eucalyptus* forests, similar to the values obtained in this study (1.0 to 0.8). Bare soil represented NDVI values close to 0, and water bodies represented negative NDVI values [71]. In this case, both sample plots had minimum values of 0.3–0.4, indicating the presence of undergrowth on the plots. The high variabilities in shrubland species induce higher variability in NDVI and facilitate the acquisition of greater variations among seasons [72].

From a holistic perspective in which the type of felling is not taken into account, but the small-scale management of a large number of small plots in a large forest area is facilitated, an accuracy of 88% was achieved. These results were similar to those obtained by [73], wherein the authors obtained a detection accuracy of approximately 90% for clear cuts in the size category above 15 ha, similar to studies that employ the Random Forest algorithm, 90% [74] and 98% [75]. In the case of this study, the methodology was specifically designed for small plots for which the average area was 0.47 ha and the maximum surface area was 3 ha; therefore, this difficulty found by other investigations was corrected. On the other hand, the Kappa index, which is used to assess concordance or reproducibility, provided a value of 0.70. Persson et al. [76], using Sentinel-2, obtained a maximum value of

0.84 and a minimum of 0.63, depending on the months used, from which it can be deduced that each season affects the analysis due to phenological variables. In our case, wherein the methodology covered 3 years and each season, it can be deduced that an optimum value was obtained. Furthermore, this study obtained similar values to Forstmaier et al. [77], where a model was used to predict *Eucalyptus* cover, with a sensitivity of 75.7% and a specificity of 95.8%.

The methodology developed herein offers a new opportunity that would improve efficient forest management as well as land-use planning. The studies focus on large plot sizes, demonstrating the great potential of using medium resolution temporal imagery [78]. However, there is no literature analyzing small plot sizes and their significance. Some studies analyzing medium-sized plots (28.4 ha) report overestimation [79]. Moreover, the methodologies are tedious and complicated, even combining three types of remote sensing data [80], and are hardly applicable by the average forest user. This study has selected the most effective and straightforward approach for small plots (<3 ha), where its control and management are more difficult. Therefore, this method can be assumed and applied for forest management by companies, owners or certifiers. Currently, in the study region, the data on the felling date came from loggers. This information is usually incomplete or wrong. Therefore, offering an automated system, which can save time and decrease error, can improve the wood handling and control system. One of the advantages of the methodology employed in this study is that it is based on pixels and the use of spectral signatures and allows the application of indices and vegetation. In this case, the index used was NDVI due to its highly demonstrated good performance. This study did not have the general objective of detecting phenological changes in the studied plots, but in this sense, NDVI demonstrated good effectiveness in correctly monitoring forested areas. This method provides the opportunity to detect problems caused by pests or other anthropogenic pressures by analyzing declines in the index value. According to the articles reviewed in which Sentinel-2 images were used, high efficiencies were obtained, but the adaptation of this method to the small plots that make up the study area was improved in this study. Therefore, this methodology would allow the creation of automation that detects felling assigned to a specific management system. In this way, resources would be saved when field data are collected through "in situ checks". In addition to speeding up the detection capacity, this process would facilitate the management system's ability to react. This application would allow monitoring to improve the achievement of forest policy objectives as well as the implementation of these detection systems in decision making for forest planning.

Regarding the required processing time, it is important to note that this time depends on the performance of the computer, the software solutions and the specific knowledge and skills of the operator. In addition, the operator's knowledge and skills in (visual and digital) image interpretation and analysis, as well as knowledge of the characteristics and conditions of the observed environment, are necessary. On the other hand, the applied procedure is considered to be fast, facilitating the rapid monitoring of a large study area consisting of multiple plots with different compositions and characteristics.

## 5. Conclusions

Based on the results obtained herein, the use of the Sentinel-2 satellite can be considered a valuable tool for forest monitoring, especially considering that these data will be routinely available for many years and that the images are frequent and free of charge.

On the other hand, many of the abrupt changes obtained in the NDVI series of a given pixel are due to cutting in most cases. NDVI could identify and quantify changes in the analyzed forest parcels. The approach presented in this study could be used in other forest stands to monitor anthropogenic changes in small forest plots. An accuracy of approximately 95% was achieved when the whole plot was cut, with an accuracy of 81% obtained for partial cuts. At the same time, the methodology used herein also facilitated

correct monitoring by the managers of these plots and effectively detected phenological changes, making it possible to assess the effects of changes in forest stands.

Finally, the presented workflow demonstrated the efficiency of the performed approach. This methodology would allow future works to improve and streamline, from an operational perspective, the processing of these images, allowing the reduction of processing time and storage capacity. This methodology is particularly useful in large research areas, in areas of multiple plots with different forest species and in areas where the property system is characterized by smallholdings.

**Author Contributions:** Conceptualization, A.L.-A., X.Á. and H.L.; Data curation, A.L.-A.; Formal analysis, A.L.-A. and J.L.R.; Funding acquisition, A.L.-A. and H.L.; Investigation, A.L.-A., X.Á. and H.L.; Methodology, A.L.-A. and J.L.R.; Project administration, H.L.; Resources, H.L.; Software, A.L.-A. and J.L.R.; Supervision, X.Á. and H.L.; Validation, A.L.-A., X.Á. and H.L.; Visualization, A.L.-A., X.Á. and J.L.R.; Writing—original draft, A.L.-A.; Writing—review and editing, X.Á., H.L. and J.L.R. All authors have read and agreed to the published version of the manuscript.

**Funding:** H.L. is the beneficiary of Xunta de Galicia grant ED431C 2020/01 for Competitive Reference Research Groups (2020–2023). In addition, this research was funded by the Conselleira de Educación, Universidade e Formación Profesional, Xunta de Galicia, España, under project R815 131H 64502 (X.A.).

**Institutional Review Board Statement:** Not applicable.

**Informed Consent Statement:** Not applicable.

**Data Availability Statement:** The authors will provide the information to whomever needs it through a request to the corresponding author.

**Acknowledgments:** The authors are grateful to ASEFOR S.L. staff for their administrative and technical support, as well as to owners and managers of the Alvariza Forest Certification Group who are committed to responsible and sustainable forest management. The authors would like to thank Carolina Acuña-Alonso for her help and contribution, as well as José Manuel Casas Mirás for his help in the final review.

**Conflicts of Interest:** The authors declare no conflict of interest. The funders had no role in the design of the study; in the collection, analyses or interpretation of data; in the writing of the manuscript or in the decision to publish the results.

## Appendix A

Data set of the 135 analyzed plots, which form the classification tree incorporated in the Rstudio software.

| Average | 25 th Percentile | 50th Percentile | 75th Percentile | 95th Percentile | Area(ha) | Perimeter (m) | Clear-Cut | Area Trend (ha) | Relation Index Descent Area—Stand Area (%) |
|---|---|---|---|---|---|---|---|---|---|
| −0.23877 | −0.31345 | −0.24974 | −0.16741 | −0.14145 | 0.063 | 189.26 | 1 | 0.06 | 95.24 |
| −0.14236 | −0.13783 | −0.13709 | −0.13009 | −0.1266 | 0.103 | 302.67 | 2 | 0.05 | 48.54 |
| −0.14921 | −0.14921 | −0.14921 | −0.14921 | −0.14921 | 0.18 | 235.27 | 2 | 0.01 | 5.56 |
| −0.2041 | −0.24096 | −0.19674 | −0.17645 | −0.1424 | 0.186 | 242.59 | 1 | 0.16 | 86.02 |
| −0.15906 | −0.17361 | −0.16287 | −0.1425 | −0.13619 | 0.194 | 348.14 | 2 | 0.07 | 36.08 |
| −0.35642 | −0.41328 | −0.35988 | −0.33433 | −0.25851 | 0.194 | 348.14 | 1 | 0.18 | 92.78 |
| −0.35454 | −0.39495 | −0.35508 | −0.31162 | −0.25826 | 0.202 | 260.5 | 1 | 0.19 | 94.06 |
| −0.15058 | −0.15058 | −0.15058 | −0.15058 | −0.15058 | 0.202 | 260.5 | 2 | 0.01 | 4.95 |
| −0.32099 | −0.36368 | −0.33208 | −0.29105 | −0.20624 | 0.204 | 387.59 | 1 | 0.2 | 98.04 |

| Average | 25 th Percentile | 50th Percentile | 75th Percentile | 95th Percentile | Area(ha) | Perimeter (m) | Clear-Cut | Area Trend (ha) | Relation Index Descent Area—Stand Area (%) |
|---|---|---|---|---|---|---|---|---|---|
| −0.17462 | −0.2024 | −0.16683 | −0.15611 | −0.13997 | 0.204 | 387.59 | 2 | 0.12 | 58.82 |
| −0.19082 | −0.2084 | −0.18672 | −0.13805 | −0.13523 | 0.204 | 387.59 | 2 | 0.05 | 24.51 |
| −0.16084 | −0.16588 | −0.16084 | −0.15579 | −0.15175 | 0.209 | 185.96 | 2 | 0.02 | 9.57 |
| −0.15295 | −0.16514 | −0.15532 | −0.13617 | −0.13004 | 0.219 | 500.02 | 2 | 0.06 | 27.4 |
| −0.34427 | −0.38731 | −0.33371 | −0.30781 | −0.27604 | 0.236 | 446.16 | 1 | 0.22 | 93.22 |
| −0.20902 | −0.20902 | −0.20902 | −0.20902 | −0.20902 | 0.236 | 446.16 | 2 | 0.01 | 4.24 |
| −0.20587 | −0.21751 | −0.20587 | −0.19422 | −0.1849 | 0.254 | 276.02 | 2 | 0.02 | 7.87 |
| −0.13207 | −0.13484 | −0.13207 | −0.12931 | −0.12709 | 0.254 | 276.02 | 2 | 0.02 | 7.87 |
| −0.35017 | −0.40293 | −0.38108 | −0.30716 | −0.18631 | 0.254 | 276.02 | 1 | 0.18 | 70.87 |
| −0.18862 | −0.18862 | −0.18862 | −0.18862 | −0.18862 | 0.254 | 276.02 | 2 | 0.01 | 3.94 |
| −0.22094 | −0.22225 | −0.22094 | −0.21962 | −0.21856 | 0.269 | 368.92 | 2 | 0.02 | 7.43 |
| −0.28922 | −0.32187 | −0.31068 | −0.25016 | −0.17622 | 0.269 | 368.92 | 1 | 0.16 | 59.48 |
| −0.12869 | −0.12995 | −0.12869 | −0.12742 | −0.12641 | 0.275 | 506.63 | 2 | 0.02 | 7.27 |
| −0.172 | −0.18233 | −0.172 | −0.16166 | −0.15339 | 0.275 | 506.63 | 2 | 0.02 | 7.27 |
| −0.12664 | −0.12664 | −0.12664 | −0.12664 | −0.12664 | 0.275 | 506.63 | 2 | 0.01 | 3.64 |
| −0.37358 | −0.42862 | −0.39838 | −0.37494 | −0.14041 | 0.275 | 506.63 | 1 | 0.19 | 69.09 |
| −0.23837 | −0.3046 | −0.26039 | −0.16894 | −0.14108 | 0.279 | 531.97 | 1 | 0.14 | 50.18 |
| −0.14238 | −0.14238 | −0.14238 | −0.14238 | −0.14238 | 0.295 | 587.95 | 2 | 0.01 | 3.39 |
| −0.14614 | −0.15365 | −0.14592 | −0.13547 | −0.13272 | 0.295 | 587.95 | 1 | 0.09 | 30.51 |
| −0.15319 | −0.16122 | −0.14954 | −0.14341 | −0.13602 | 0.364 | 252.12 | 2 | 0.07 | 19.23 |
| −0.14546 | −0.14617 | −0.14546 | −0.14476 | −0.14419 | 0.368 | 253.8 | 2 | 0.02 | 5.43 |
| −0.15973 | −0.16669 | −0.15339 | −0.14643 | −0.1411 | 0.38 | 270.81 | 2 | 0.04 | 10.53 |
| −0.4217 | −0.45081 | −0.42437 | −0.40781 | −0.36641 | 0.38 | 270.81 | 1 | 0.39 | 102.63 |
| −0.30046 | −0.33721 | −0.30511 | −0.27612 | −0.18231 | 0.383 | 248.68 | 1 | 0.29 | 75.72 |
| −0.14091 | −0.14793 | −0.13359 | −0.13088 | −0.13059 | 0.383 | 248.68 | 2 | 0.05 | 13.05 |
| −0.15896 | −0.17657 | −0.14134 | −0.13374 | −0.12871 | 0.39 | 309.34 | 2 | 0.07 | 17.95 |
| −0.14411 | −0.14516 | −0.14411 | −0.14306 | −0.14222 | 0.39 | 309.34 | 2 | 0.02 | 5.13 |
| −0.15863 | −0.17297 | −0.1497 | −0.14728 | −0.14414 | 0.39 | 309.34 | 1 | 0.05 | 12.82 |
| −0.16014 | −0.16014 | −0.16014 | −0.16014 | −0.16014 | 0.402 | 408.01 | 2 | 0.01 | 2.49 |
| −0.13443 | −0.13919 | −0.1341 | −0.12933 | −0.12783 | 0.402 | 408.01 | 2 | 0.04 | 9.95 |
| −0.15494 | −0.16401 | −0.15768 | −0.1472 | −0.13283 | 0.413 | 457.7 | 2 | 0.21 | 50.85 |
| −0.24233 | −0.29505 | −0.16709 | −0.14485 | −0.12801 | 0.413 | 457.7 | 1 | 0.31 | 75.06 |
| −0.27044 | −0.32025 | −0.29086 | −0.23921 | −0.13733 | 0.413 | 457.7 | 2 | 0.16 | 38.74 |
| −0.35753 | −0.39893 | −0.35753 | −0.31612 | −0.28299 | 0.415 | 689.22 | 2 | 0.02 | 4.82 |
| −0.234 | −0.27463 | −0.20422 | −0.15621 | −0.13366 | 0.415 | 689.22 | 2 | 0.36 | 86.75 |
| −0.35063 | −0.44136 | −0.35593 | −0.26473 | −0.17073 | 0.415 | 332.78 | 1 | 0.39 | 93.98 |
| −0.19055 | −0.20578 | −0.18382 | −0.16341 | −0.14216 | 0.453 | 320.27 | 2 | 0.11 | 24.28 |
| −0.20862 | −0.24305 | −0.21826 | −0.16955 | −0.14365 | 0.453 | 320.27 | 1 | 0.27 | 59.6 |
| −0.20522 | −0.22355 | −0.19334 | −0.1765 | −0.15971 | 0.462 | 502.26 | 2 | 0.07 | 15.15 |
| −0.15046 | −0.15215 | −0.15051 | −0.1488 | −0.14743 | 0.462 | 502.26 | 2 | 0.03 | 6.49 |
| −0.30753 | −0.38042 | −0.33914 | −0.21204 | −0.14985 | 0.462 | 502.26 | 1 | 0.29 | 62.77 |
| −0.3824 | −0.41636 | −0.40474 | −0.38373 | −0.22455 | 0.463 | 315.55 | 1 | 0.45 | 97.19 |
| −0.198 | −0.19826 | −0.198 | −0.19775 | −0.19754 | 0.463 | 315.55 | 2 | 0.02 | 4.32 |
| −0.15278 | −0.15278 | −0.15278 | −0.15278 | −0.15278 | 0.492 | 423.09 | 2 | 0.01 | 2.03 |

| Average | 25 th Percentile | 50th Percentile | 75th Percentile | 95th Percentile | Area(ha) | Perimeter (m) | Clear-Cut | Area Trend (ha) | Relation Index Descent Area—Stand Area (%) |
|---|---|---|---|---|---|---|---|---|---|
| −0.15099 | −0.16126 | −0.14946 | −0.13644 | −0.13032 | 0.492 | 423.09 | 2 | 0.25 | 50.81 |
| −0.29542 | −0.34644 | −0.29376 | −0.27071 | −0.191 | 0.492 | 423.09 | 1 | 0.46 | 93.5 |
| −0.16454 | −0.16454 | −0.16454 | −0.16454 | −0.16454 | 0.499 | 302.34 | 2 | 0.01 | 2 |
| −0.21635 | −0.26086 | −0.22876 | −0.16883 | −0.14041 | 0.522 | 302.47 | 1 | 0.35 | 67.05 |
| −0.16017 | −0.17618 | −0.16173 | −0.14332 | −0.13934 | 0.534 | 772.16 | 2 | 0.05 | 9.36 |
| −0.29322 | −0.32089 | −0.29705 | −0.26273 | −0.20361 | 0.551 | 433.42 | 1 | 0.49 | 88.93 |
| −0.13408 | −0.13488 | −0.13408 | −0.13327 | −0.13263 | 0.551 | 433.42 | 2 | 0.02 | 3.63 |
| −0.16567 | −0.18474 | −0.16381 | −0.14497 | −0.13705 | 0.577 | 504.69 | 2 | 0.1 | 17.33 |
| −0.24706 | −0.32224 | −0.25198 | −0.17758 | −0.1348 | 0.577 | 504.69 | 1 | 0.26 | 45.06 |
| −0.20405 | −0.25438 | −0.162 | −0.14291 | −0.12685 | 0.587 | 523.69 | 1 | 0.25 | 42.59 |
| −0.12965 | −0.12965 | −0.12965 | −0.12965 | −0.12965 | 0.587 | 523.69 | 2 | 0.01 | 1.7 |
| −0.21846 | −0.2488 | −0.23224 | −0.17683 | −0.16231 | 0.605 | 389.32 | 1 | 0.07 | 11.57 |
| −0.15958 | −0.15958 | −0.15958 | −0.15958 | −0.15958 | 0.605 | 389.32 | 2 | 0.01 | 1.65 |
| −0.16518 | −0.19373 | −0.15261 | −0.13244 | −0.12864 | 0.611 | 472.01 | 2 | 0.05 | 8.18 |
| −0.1471 | −0.14699 | −0.14561 | −0.14256 | −0.13308 | 0.611 | 472.01 | 2 | 0.09 | 14.73 |
| −0.15872 | −0.17026 | −0.15525 | −0.14544 | −0.13759 | 0.611 | 472.01 | 2 | 0.03 | 4.91 |
| −0.36968 | −0.41931 | −0.3967 | −0.33556 | −0.2194 | 0.611 | 472.01 | 1 | 0.29 | 47.46 |
| −0.16747 | −0.18176 | −0.17217 | −0.14669 | −0.13453 | 0.611 | 472.01 | 2 | 0.12 | 19.64 |
| −0.12948 | −0.12948 | −0.12948 | −0.12948 | −0.12948 | 0.614 | 315.79 | 2 | 0.01 | 1.63 |
| −0.22906 | −0.25549 | −0.22906 | −0.20262 | −0.18148 | 0.614 | 315.79 | 1 | 0.02 | 3.26 |
| −0.16531 | −0.17928 | −0.16814 | −0.14702 | −0.13407 | 0.614 | 315.79 | 2 | 0.1 | 16.29 |
| −0.14818 | −0.15221 | −0.14267 | −0.13863 | −0.1354 | 0.614 | 315.79 | 2 | 0.04 | 6.51 |
| −0.19836 | −0.19946 | −0.19836 | −0.19727 | −0.19639 | 0.619 | 431.28 | 2 | 0.02 | 3.23 |
| −0.24374 | −0.3085 | −0.23594 | −0.18562 | −0.13743 | 0.619 | 431.28 | 1 | 0.46 | 74.31 |
| −0.14678 | −0.1504 | −0.14678 | −0.14315 | −0.14025 | 0.619 | 431.28 | 2 | 0.02 | 3.23 |
| −0.1378 | −0.13947 | −0.13594 | −0.12972 | −0.12752 | 0.619 | 431.28 | 2 | 0.05 | 8.08 |
| −0.15093 | −0.15295 | −0.15093 | −0.1489 | −0.14728 | 0.619 | 431.28 | 2 | 0.02 | 3.23 |
| −0.30772 | −0.37153 | −0.31911 | −0.23704 | −0.17061 | 0.629 | 345.62 | 1 | 0.6 | 95.39 |
| −0.16679 | −0.18572 | −0.16075 | −0.13936 | −0.13491 | 0.629 | 345.62 | 2 | 0.08 | 12.72 |
| −0.14661 | −0.14661 | −0.14661 | −0.14661 | −0.14661 | 0.638 | 683.38 | 2 | 0.01 | 1.57 |
| −0.14974 | −0.17166 | −0.1389 | −0.1327 | −0.13226 | 0.638 | 683.38 | 2 | 0.05 | 7.84 |
| −0.21601 | −0.24116 | −0.22003 | −0.19058 | −0.15764 | 0.638 | 683.38 | 1 | 0.58 | 90.91 |
| −0.20602 | −0.22973 | −0.19779 | −0.17772 | −0.16139 | 0.668 | 343.15 | 1 | 0.12 | 17.96 |
| −0.17343 | −0.2169 | −0.15745 | −0.14348 | −0.12748 | 0.668 | 343.15 | 2 | 0.1 | 14.97 |
| −0.15988 | −0.17447 | −0.14973 | −0.1443 | −0.13853 | 0.668 | 343.15 | 2 | 0.08 | 11.98 |
| −0.17213 | −0.1877 | −0.18008 | −0.16451 | −0.13819 | 0.668 | 343.15 | 2 | 0.04 | 5.99 |
| −0.32665 | −0.3704 | −0.32236 | −0.29145 | −0.2461 | 0.692 | 435.44 | 1 | 0.72 | 104.05 |
| −0.22025 | −0.24663 | −0.22781 | −0.20284 | −0.14171 | 0.692 | 435.44 | 2 | 0.42 | 60.69 |
| −0.13173 | −0.13173 | −0.13173 | −0.13173 | −0.13173 | 0.707 | 360.16 | 2 | 0.01 | 1.41 |
| −0.41584 | −0.45852 | −0.44072 | −0.41877 | −0.21535 | 0.707 | 360.16 | 1 | 0.65 | 91.94 |
| −0.15118 | −0.1559 | −0.14022 | −0.13288 | −0.13137 | 0.71 | 412.36 | 2 | 0.05 | 7.04 |
| −0.33678 | −0.44056 | −0.32327 | −0.2454 | −0.15562 | 0.71 | 412.36 | 1 | 0.61 | 85.92 |
| −0.12591 | −0.12591 | −0.12591 | −0.12591 | −0.12591 | 0.715 | 359.9 | 2 | 0.01 | 1.4 |
| −0.14878 | −0.1542 | −0.14878 | −0.14335 | −0.13901 | 0.715 | 359.9 | 2 | 0.02 | 2.8 |

| Average | 25 th Percentile | 50th Percentile | 75th Percentile | 95th Percentile | Area(ha) | Perimeter (m) | Clear-Cut | Area Trend (ha) | Relation Index Descent Area—Stand Area (%) |
|---|---|---|---|---|---|---|---|---|---|
| −0.23194 | −0.26184 | −0.23153 | −0.21037 | −0.15156 | 0.715 | 359.9 | 1 | 0.44 | 61.54 |
| −0.14305 | −0.14305 | −0.14305 | −0.14305 | −0.14305 | 0.715 | 359.9 | 2 | 0.01 | 1.4 |
| −0.33401 | −0.40794 | −0.35568 | −0.278 | −0.15586 | 0.716 | 432.83 | 1 | 0.54 | 75.42 |
| −0.33509 | −0.41056 | −0.36944 | −0.26877 | −0.16603 | 0.746 | 462.69 | 1 | 0.51 | 68.36 |
| −0.17877 | −0.2117 | −0.1808 | −0.14591 | −0.1348 | 0.746 | 462.69 | 2 | 0.07 | 9.38 |
| −0.26115 | −0.31091 | −0.27793 | −0.21521 | −0.13916 | 0.748 | 544.1 | 1 | 0.65 | 86.9 |
| −0.19861 | −0.21322 | −0.19861 | −0.18401 | −0.17232 | 0.748 | 544.1 | 2 | 0.02 | 2.67 |
| −0.20047 | −0.23462 | −0.19901 | −0.1694 | −0.1352 | 0.748 | 544.1 | 2 | 0.3 | 40.11 |
| −0.30514 | −0.34035 | −0.31851 | −0.28212 | −0.20341 | 0.765 | 444.89 | 1 | 0.71 | 92.81 |
| −0.13666 | −0.13892 | −0.13546 | −0.1338 | −0.13247 | 0.765 | 444.89 | 2 | 0.03 | 3.92 |
| −0.38992 | −0.48775 | −0.41262 | −0.31138 | −0.16222 | 0.769 | 411.84 | 1 | 0.69 | 89.73 |
| −0.14309 | −0.15088 | −0.14352 | −0.13573 | −0.12897 | 0.769 | 411.84 | 2 | 0.04 | 5.2 |
| −0.16355 | −0.16911 | −0.16717 | −0.1598 | −0.1539 | 0.769 | 411.84 | 2 | 0.03 | 3.9 |
| −0.40813 | −0.46684 | −0.43483 | −0.39507 | −0.22932 | 0.773 | 362.02 | 1 | 0.7 | 90.56 |
| −0.16374 | −0.17565 | −0.17277 | −0.15892 | −0.13462 | 0.773 | 362.02 | 2 | 0.09 | 11.64 |
| −0.17646 | −0.19277 | −0.14431 | −0.14408 | −0.1439 | 0.773 | 362.02 | 2 | 0.03 | 3.88 |
| −0.19216 | −0.20802 | −0.18749 | −0.16387 | −0.14497 | 0.773 | 362.02 | 2 | 0.11 | 14.23 |
| −0.32543 | −0.40899 | −0.3326 | −0.24832 | −0.16265 | 0.791 | 462.16 | 1 | 0.64 | 80.91 |
| −0.17959 | −0.19274 | −0.18529 | −0.15754 | −0.13686 | 0.791 | 462.16 | 2 | 0.17 | 21.49 |
| −0.17452 | −0.20626 | −0.15164 | −0.13981 | −0.13626 | 0.791 | 462.16 | 2 | 0.11 | 13.91 |
| −0.13942 | −0.14315 | −0.13962 | −0.13579 | −0.13273 | 0.791 | 462.16 | 2 | 0.03 | 3.79 |
| −0.15858 | −0.15858 | −0.15858 | −0.15858 | −0.15858 | 0.791 | 462.16 | 2 | 0.01 | 1.26 |
| −0.16151 | −0.17943 | −0.16919 | −0.14743 | −0.13002 | 0.791 | 462.16 | 2 | 0.03 | 3.79 |
| −0.17818 | −0.2009 | −0.17053 | −0.15725 | −0.14602 | 0.795 | 356.79 | 2 | 0.07 | 8.81 |
| −0.18881 | −0.21064 | −0.17243 | −0.15773 | −0.13437 | 0.795 | 356.79 | 2 | 0.1 | 12.58 |
| −0.13971 | −0.14571 | −0.14383 | −0.13577 | −0.12932 | 0.821 | 361.93 | 2 | 0.03 | 3.65 |
| −0.2252 | −0.26877 | −0.21795 | −0.18538 | −0.135 | 0.821 | 361.93 | 1 | 0.68 | 82.83 |
| −0.14697 | −0.15684 | −0.12913 | −0.12819 | −0.12744 | 0.821 | 361.93 | 2 | 0.03 | 3.65 |
| −0.14155 | −0.14333 | −0.14155 | −0.13976 | −0.13834 | 0.821 | 361.93 | 2 | 0.02 | 2.44 |
| −0.23558 | −0.27408 | −0.23837 | −0.19542 | −0.14678 | 0.823 | 452.19 | 1 | 0.61 | 74.12 |
| −0.13668 | −0.14003 | −0.13807 | −0.1336 | −0.1304 | 0.823 | 452.19 | 2 | 0.07 | 8.51 |
| −0.27042 | −0.30374 | −0.27551 | −0.23732 | −0.18075 | 0.861 | 440.34 | 1 | 0.76 | 88.27 |
| −0.12693 | −0.12693 | −0.12693 | −0.12693 | −0.12693 | 0.876 | 390.47 | 2 | 0.01 | 1.14 |
| −0.13706 | −0.13706 | −0.13706 | −0.13706 | −0.13706 | 0.876 | 390.47 | 2 | 0.01 | 1.14 |
| −0.32596 | −0.4188 | −0.37254 | −0.23004 | −0.1688 | 0.876 | 390.47 | 1 | 0.44 | 50.23 |
| −0.20173 | −0.21917 | −0.19154 | −0.1792 | −0.16933 | 0.886 | 467.32 | 2 | 0.03 | 3.39 |
| −0.35299 | −0.45323 | −0.35731 | −0.26468 | −0.16272 | 0.886 | 467.32 | 1 | 0.68 | 76.75 |
| −0.18605 | −0.22464 | −0.14717 | −0.13747 | −0.1317 | 0.886 | 467.32 | 2 | 0.11 | 12.42 |
| −0.19481 | −0.23776 | −0.19167 | −0.15523 | −0.13802 | 0.886 | 467.32 | 2 | 0.23 | 25.96 |
| −0.1572 | −0.1572 | −0.1572 | −0.1572 | −0.1572 | 0.886 | 467.32 | 2 | 0.01 | 1.13 |
| −0.30398 | −0.34898 | −0.31487 | −0.27458 | −0.18144 | 0.9 | 455.27 | 1 | 0.83 | 92.22 |
| −0.2989 | −0.34883 | −0.29243 | −0.24718 | −0.19363 | 0.9 | 455.27 | 2 | 0.42 | 46.67 |
| −0.12757 | −0.12757 | −0.12757 | −0.12757 | −0.12757 | 0.9 | 455.27 | 2 | 0.01 | 1.11 |
| −0.22628 | −0.24534 | −0.23713 | −0.21807 | −0.18761 | 0.9 | 455.27 | 2 | 0.04 | 4.44 |

| Average | 25 th Percentile | 50th Percentile | 75th Percentile | 95th Percentile | Area(ha) | Perimeter (m) | Clear-Cut | Area Trend (ha) | Relation Index Descent Area—Stand Area (%) |
|---|---|---|---|---|---|---|---|---|---|
| −0.27436 | −0.34514 | −0.26262 | −0.2139 | −0.14133 | 0.908 | 511.18 | 1 | 0.83 | 91.41 |
| −0.16442 | −0.17681 | −0.16442 | −0.15204 | −0.14213 | 0.908 | 511.18 | 2 | 0.02 | 2.2 |
| −0.16197 | −0.1764 | −0.16096 | −0.13763 | −0.12823 | 0.913 | 448.16 | 2 | 0.29 | 31.76 |
| −0.15498 | −0.16496 | −0.16335 | −0.14919 | −0.13786 | 0.913 | 448.16 | 2 | 0.03 | 3.29 |
| −0.41859 | −0.46664 | −0.45126 | −0.40822 | −0.28365 | 0.918 | 458.24 | 2 | 0.23 | 25.05 |
| −0.14989 | −0.15397 | −0.14989 | −0.1458 | −0.14253 | 0.918 | 458.24 | 2 | 0.02 | 2.18 |
| −0.17522 | −0.19846 | −0.17145 | −0.15766 | −0.14598 | 0.918 | 458.24 | 2 | 0.08 | 8.71 |
| −0.2094 | −0.23321 | −0.22906 | −0.19543 | −0.16852 | 0.918 | 458.24 | 2 | 0.03 | 3.27 |
| −0.17523 | −0.19103 | −0.1748 | −0.15665 | −0.13807 | 0.918 | 458.24 | 2 | 0.27 | 29.41 |
| −0.13903 | −0.14348 | −0.13722 | −0.13368 | −0.13084 | 0.918 | 458.24 | 2 | 0.03 | 3.27 |
| −0.19098 | −0.20797 | −0.19111 | −0.17328 | −0.1395 | 0.918 | 458.24 | 1 | 0.24 | 26.14 |
| −0.15849 | −0.1713 | −0.15315 | −0.14302 | −0.13492 | 0.943 | 710.25 | 2 | 0.03 | 3.18 |
| −0.14374 | −0.14787 | −0.14374 | −0.13961 | −0.13631 | 0.943 | 710.25 | 2 | 0.02 | 2.12 |
| −0.33153 | −0.39665 | −0.32818 | −0.28573 | −0.18487 | 0.943 | 710.25 | 1 | 0.55 | 58.32 |
| −0.15729 | −0.16219 | −0.14834 | −0.12958 | −0.12601 | 0.943 | 710.25 | 2 | 0.11 | 11.66 |
| −0.15294 | −0.14484 | −0.13384 | −0.12997 | −0.12704 | 0.943 | 710.25 | 2 | 0.11 | 11.66 |
| −0.14583 | −0.14583 | −0.14583 | −0.14583 | −0.14583 | 0.962 | 418.73 | 2 | 0.01 | 1.04 |
| −0.40231 | −0.47654 | −0.44329 | −0.37441 | −0.1743 | 0.962 | 418.73 | 1 | 0.78 | 81.08 |
| −0.12961 | −0.12961 | −0.12961 | −0.12961 | −0.12961 | 0.962 | 418.73 | 2 | 0.01 | 1.04 |
| −0.22111 | −0.23154 | −0.22111 | −0.21067 | −0.20233 | 0.962 | 418.73 | 2 | 0.02 | 2.08 |
| −0.14114 | −0.14847 | −0.14137 | −0.13182 | −0.12788 | 0.962 | 418.73 | 2 | 0.14 | 14.55 |
| −0.27336 | −0.31871 | −0.2918 | −0.25483 | −0.15325 | 0.962 | 418.73 | 2 | 0.21 | 21.83 |
| −0.38266 | −0.41375 | −0.38909 | −0.37212 | −0.25544 | 0.984 | 689.22 | 1 | 0.99 | 100.61 |
| −0.14676 | −0.14676 | −0.14676 | −0.14676 | −0.14676 | 0.984 | 689.22 | 2 | 0.01 | 1.02 |
| −0.15851 | −0.16857 | −0.15851 | −0.14844 | −0.14038 | 0.984 | 689.22 | 2 | 0.02 | 2.03 |
| −0.17792 | −0.17792 | −0.17792 | −0.17792 | −0.17792 | 0.984 | 689.22 | 2 | 0.01 | 1.02 |
| −0.14446 | −0.14621 | −0.14184 | −0.13858 | −0.1317 | 0.996 | 477.71 | 2 | 0.16 | 16.06 |
| −0.35946 | −0.41441 | −0.35596 | −0.31942 | −0.28651 | 0.996 | 477.71 | 1 | 0.05 | 5.02 |
| −0.34873 | −0.43569 | −0.39303 | −0.27798 | −0.14455 | 1.006 | 638.79 | 1 | 0.81 | 80.52 |
| −0.12544 | −0.12544 | −0.12544 | −0.12544 | −0.12544 | 1.006 | 638.79 | 2 | 0.01 | 0.99 |
| −0.18772 | −0.18772 | −0.18772 | −0.18772 | −0.18772 | 1.006 | 638.79 | 2 | 0.01 | 0.99 |
| −0.18438 | −0.20876 | −0.18311 | −0.15051 | −0.12933 | 1.006 | 638.79 | 2 | 0.19 | 18.89 |
| −0.16488 | −0.1787 | −0.15481 | −0.1347 | −0.13071 | 1.006 | 638.79 | 2 | 0.1 | 9.94 |
| −0.17401 | −0.19803 | −0.17732 | −0.14536 | −0.13059 | 1.006 | 638.79 | 2 | 0.23 | 22.86 |
| −0.14198 | −0.14198 | −0.14198 | −0.14198 | −0.14198 | 1.006 | 638.79 | 2 | 0.01 | 0.99 |
| −0.1569 | −0.16131 | −0.15245 | −0.14804 | −0.13986 | 1.006 | 638.79 | 2 | 0.04 | 3.98 |
| −0.14617 | −0.1543 | −0.14617 | −0.13804 | −0.13154 | 1.006 | 638.79 | 2 | 0.02 | 1.99 |
| −0.15406 | −0.15406 | −0.15406 | −0.15406 | −0.15406 | 1.008 | 400.03 | 2 | 0.01 | 0.99 |
| −0.18067 | −0.18067 | −0.18067 | −0.18067 | −0.18067 | 1.008 | 400.03 | 2 | 0.01 | 0.99 |
| −0.25052 | −0.32763 | −0.24137 | −0.1525 | −0.12659 | 1.008 | 400.03 | 1 | 0.21 | 20.83 |
| −0.14487 | −0.1482 | −0.13723 | −0.13482 | −0.12823 | 1.008 | 400.03 | 2 | 0.09 | 8.93 |
| −0.16071 | −0.16336 | −0.15853 | −0.15588 | −0.15476 | 1.008 | 400.03 | 2 | 0.04 | 3.97 |
| −0.14141 | −0.14847 | −0.12867 | −0.12798 | −0.12742 | 1.008 | 400.03 | 2 | 0.03 | 2.98 |
| −0.14333 | −0.1456 | −0.14389 | −0.14135 | −0.13931 | 1.017 | 427.96 | 2 | 0.03 | 2.95 |

| Average | 25 th Percentile | 50th Percentile | 75th Percentile | 95th Percentile | Area(ha) | Perimeter (m) | Clear-Cut | Area Trend (ha) | Relation Index Descent Area—Stand Area (%) |
|---|---|---|---|---|---|---|---|---|---|
| −0.2278 | −0.26249 | −0.23562 | −0.18829 | −0.15082 | 1.017 | 427.96 | 1 | 0.61 | 59.98 |
| −0.15469 | −0.16526 | −0.152 | −0.138 | −0.13083 | 1.017 | 427.96 | 2 | 0.24 | 23.6 |
| −0.14555 | −0.15868 | −0.14004 | −0.13363 | −0.13076 | 1.017 | 427.96 | 2 | 0.08 | 7.87 |
| −0.1718 | −0.18515 | −0.17864 | −0.15001 | −0.13987 | 1.017 | 427.96 | 2 | 0.08 | 7.87 |
| −0.33715 | −0.40961 | −0.32745 | −0.26418 | −0.15354 | 1.019 | 467.94 | 1 | 0.98 | 96.17 |
| −0.15995 | −0.16928 | −0.15396 | −0.14269 | −0.13495 | 1.019 | 467.94 | 2 | 0.1 | 9.81 |
| −0.15515 | −0.16715 | −0.15093 | −0.14012 | −0.1302 | 1.026 | 403.8 | 2 | 0.11 | 10.72 |
| −0.14079 | −0.14079 | −0.14079 | −0.14079 | −0.14079 | 1.026 | 403.8 | 2 | 0.01 | 0.97 |
| −0.14064 | −0.14064 | −0.14064 | −0.14064 | −0.14064 | 1.026 | 403.8 | 2 | 0.01 | 0.97 |
| −0.29391 | −0.36428 | −0.28989 | −0.22696 | −0.15581 | 1.038 | 579.92 | 1 | 0.28 | 26.97 |
| −0.23074 | −0.25826 | −0.22853 | −0.19238 | −0.14115 | 1.048 | 529.15 | 1 | 0.84 | 80.15 |
| −0.12824 | −0.12824 | −0.12824 | −0.12824 | −0.12824 | 1.048 | 529.15 | 2 | 0.01 | 0.95 |
| −0.21239 | −0.25105 | −0.21075 | −0.17251 | −0.14032 | 1.061 | 474.72 | 1 | 0.93 | 87.65 |
| −0.2477 | −0.3118 | −0.245 | −0.19237 | −0.1346 | 1.075 | 450.52 | 1 | 0.59 | 54.88 |
| −0.13342 | −0.13536 | −0.13342 | −0.13149 | −0.12994 | 1.075 | 450.52 | 2 | 0.02 | 1.86 |
| −0.15088 | −0.157 | −0.14437 | −0.13798 | −0.12863 | 1.075 | 450.52 | 2 | 0.33 | 30.7 |
| −0.13013 | −0.13013 | −0.13013 | −0.13013 | −0.13013 | 1.075 | 450.52 | 2 | 0.01 | 0.93 |
| −0.20505 | −0.26001 | −0.19054 | −0.15428 | −0.1381 | 1.075 | 450.52 | 2 | 0.1 | 9.3 |
| −0.16455 | −0.16455 | −0.16455 | −0.16455 | −0.16455 | 1.089 | 474.39 | 2 | 0.01 | 0.92 |
| −0.48956 | −0.53617 | −0.51696 | −0.48819 | −0.26834 | 1.089 | 474.39 | 1 | 1.05 | 96.42 |
| −0.2871 | −0.34301 | −0.29311 | −0.2297 | −0.14911 | 1.097 | 761.25 | 1 | 0.98 | 89.33 |
| −0.13857 | −0.14033 | −0.13857 | −0.13682 | −0.13541 | 1.097 | 761.25 | 2 | 0.02 | 1.82 |
| −0.24232 | −0.28211 | −0.2513 | −0.21288 | −0.15023 | 1.102 | 466.63 | 2 | 0.99 | 89.84 |
| −0.2712 | −0.31745 | −0.269 | −0.23001 | −0.15991 | 1.102 | 466.63 | 1 | 0.99 | 89.84 |
| −0.14535 | −0.14535 | −0.14535 | −0.14535 | −0.14535 | 1.102 | 466.63 | 2 | 0.01 | 0.91 |
| −0.15925 | −0.15925 | −0.15925 | −0.15925 | −0.15925 | 1.102 | 466.63 | 2 | 0.01 | 0.91 |
| −0.13492 | −0.13495 | −0.13492 | −0.13488 | −0.13486 | 1.102 | 466.63 | 2 | 0.02 | 1.81 |
| −0.13239 | −0.13572 | −0.13241 | −0.12907 | −0.12639 | 1.108 | 488.13 | 2 | 0.03 | 2.71 |
| −0.48828 | −0.5628 | −0.52159 | −0.45329 | −0.22244 | 1.108 | 488.13 | 1 | 1.09 | 98.38 |
| −0.13775 | −0.13775 | −0.13775 | −0.13775 | −0.13775 | 1.117 | 752 | 2 | 0.01 | 0.9 |
| −0.14683 | −0.15524 | −0.14683 | −0.13841 | −0.13168 | 1.117 | 752 | 2 | 0.02 | 1.79 |
| −0.33481 | −0.36989 | −0.34567 | −0.31139 | −0.22684 | 1.117 | 752 | 1 | 1.13 | 101.16 |
| −0.13514 | −0.13514 | −0.13514 | −0.13514 | −0.13514 | 1.117 | 752 | 2 | 0.01 | 0.9 |
| −0.15641 | −0.16981 | −0.16895 | −0.14929 | −0.13355 | 1.177 | 448.82 | 2 | 0.03 | 2.55 |
| −0.15435 | −0.15333 | −0.14336 | −0.12951 | −0.12811 | 1.177 | 448.82 | 2 | 0.05 | 4.25 |
| −0.22617 | −0.26021 | −0.22781 | −0.19598 | −0.13337 | 1.177 | 448.82 | 1 | 1.06 | 90.06 |
| −0.1503 | −0.1503 | −0.1503 | −0.1503 | −0.1503 | 1.177 | 448.82 | 2 | 0.01 | 0.85 |
| −0.16354 | −0.18519 | −0.16598 | −0.1495 | −0.13137 | 1.177 | 448.82 | 2 | 0.05 | 4.25 |
| −0.21924 | −0.31437 | −0.17139 | −0.15599 | −0.13296 | 1.177 | 448.82 | 2 | 0.1 | 8.5 |
| −0.1489 | −0.16752 | −0.13855 | −0.12969 | −0.12645 | 1.244 | 507.63 | 2 | 0.36 | 28.94 |
| −0.15502 | −0.163 | −0.15071 | −0.14086 | −0.13324 | 1.244 | 507.63 | 2 | 0.18 | 14.47 |
| −0.13694 | −0.13809 | −0.13694 | −0.1358 | −0.13488 | 1.247 | 501.95 | 2 | 0.02 | 1.6 |
| −0.42585 | −0.47734 | −0.42778 | −0.39049 | −0.30692 | 1.247 | 501.95 | 1 | 1.22 | 97.83 |
| −0.29315 | −0.33941 | −0.30152 | −0.24793 | −0.16824 | 1.253 | 654.51 | 2 | 0.85 | 67.84 |

| Average | 25 th Percentile | 50th Percentile | 75th Percentile | 95th Percentile | Area(ha) | Perimeter (m) | Clear-Cut | Area Trend (ha) | Relation Index Descent Area—Stand Area (%) |
|---|---|---|---|---|---|---|---|---|---|
| −0.15036 | −0.15588 | −0.15036 | −0.14483 | −0.14041 | 1.253 | 654.51 | 2 | 0.02 | 1.6 |
| −0.20444 | −0.24331 | −0.19627 | −0.16347 | −0.13641 | 1.253 | 654.51 | 1 | 0.59 | 47.09 |
| −0.14777 | −0.15588 | −0.14586 | −0.13894 | −0.13109 | 1.253 | 654.51 | 2 | 0.06 | 4.79 |
| −0.14255 | −0.14455 | −0.13985 | −0.13921 | −0.13869 | 1.253 | 654.51 | 2 | 0.03 | 2.39 |
| −0.16998 | −0.17876 | −0.1688 | −0.15795 | −0.13982 | 1.254 | 595.13 | 2 | 0.06 | 4.78 |
| −0.24141 | −0.28362 | −0.23381 | −0.18437 | −0.1402 | 1.254 | 595.13 | 1 | 0.35 | 27.91 |
| −0.12587 | −0.12587 | −0.12587 | −0.12587 | −0.12587 | 1.254 | 595.13 | 2 | 0.01 | 0.8 |
| −0.1388 | −0.1388 | −0.1388 | −0.1388 | −0.1388 | 1.254 | 595.13 | 2 | 0.01 | 0.8 |
| −0.15553 | −0.16958 | −0.16827 | −0.14785 | −0.13151 | 1.254 | 595.13 | 2 | 0.03 | 2.39 |
| −0.26 | −0.31648 | −0.27511 | −0.1715 | −0.13074 | 1.298 | 454.85 | 1 | 0.94 | 72.42 |
| −0.13227 | −0.13227 | −0.13227 | −0.13227 | −0.13227 | 1.298 | 454.85 | 2 | 0.01 | 0.77 |
| −0.14447 | −0.15416 | −0.14141 | −0.13172 | −0.12889 | 1.298 | 454.85 | 2 | 0.04 | 3.08 |
| −0.14907 | −0.16029 | −0.14752 | −0.13707 | −0.12871 | 1.307 | 482.94 | 2 | 0.03 | 2.3 |
| −0.15046 | −0.16548 | −0.14698 | −0.12761 | −0.12672 | 1.307 | 482.94 | 2 | 0.06 | 4.59 |
| −0.20973 | −0.23732 | −0.19871 | −0.18646 | −0.14718 | 1.307 | 482.94 | 2 | 0.3 | 22.95 |
| −0.3389 | −0.4469 | −0.39513 | −0.21973 | −0.14023 | 1.307 | 482.94 | 1 | 0.82 | 62.74 |
| −0.2197 | −0.2525 | −0.21202 | −0.1867 | −0.13882 | 1.386 | 477.89 | 1 | 0.99 | 71.43 |
| −0.14198 | −0.14198 | −0.14198 | −0.14198 | −0.14198 | 1.386 | 477.89 | 2 | 0.01 | 0.72 |
| −0.17832 | −0.1791 | −0.17832 | −0.17755 | −0.17693 | 1.4 | 905.49 | 2 | 0.02 | 1.43 |
| −0.34794 | −0.40323 | −0.38166 | −0.31765 | −0.18922 | 1.4 | 905.49 | 1 | 1.41 | 100.71 |
| −0.12593 | −0.12593 | −0.12593 | −0.12593 | −0.12593 | 1.4 | 905.49 | 2 | 0.01 | 0.71 |
| −0.14346 | −0.15575 | −0.13949 | −0.13506 | −0.12854 | 1.414 | 604.32 | 2 | 0.05 | 3.54 |
| −0.14042 | −0.14652 | −0.13546 | −0.12928 | −0.12783 | 1.414 | 604.32 | 2 | 0.06 | 4.24 |
| −0.15487 | −0.16991 | −0.15149 | −0.14325 | −0.1298 | 1.414 | 604.32 | 2 | 0.36 | 25.46 |
| −0.14767 | −0.14767 | −0.14767 | −0.14767 | −0.14767 | 1.459 | 606.09 | 2 | 0.01 | 0.69 |
| −0.1387 | −0.14234 | −0.13889 | −0.13525 | −0.12904 | 1.459 | 639.78 | 2 | 0.04 | 2.74 |
| −0.14667 | −0.15867 | −0.14224 | −0.13062 | −0.12641 | 1.459 | 606.09 | 2 | 0.24 | 16.45 |
| −0.14097 | −0.14097 | −0.14097 | −0.14097 | −0.14097 | 1.459 | 639.78 | 2 | 0.01 | 0.69 |
| −0.12671 | −0.12671 | −0.12671 | −0.12671 | −0.12671 | 1.459 | 606.09 | 2 | 0.01 | 0.69 |
| −0.15664 | −0.16191 | −0.15274 | −0.13665 | −0.13007 | 1.459 | 606.09 | 2 | 0.14 | 9.6 |
| −0.20327 | −0.23658 | −0.19115 | −0.16263 | −0.13083 | 1.459 | 606.09 | 1 | 0.21 | 14.39 |
| −0.34267 | −0.41808 | −0.37392 | −0.27596 | −0.1836 | 1.459 | 639.78 | 1 | 1.3 | 89.1 |
| −0.35398 | −0.41715 | −0.37863 | −0.3037 | −0.20019 | 1.476 | 683.82 | 1 | 1.47 | 99.59 |
| −0.13755 | −0.14252 | −0.14027 | −0.13394 | −0.12887 | 1.476 | 683.82 | 2 | 0.03 | 2.03 |
| −0.14533 | −0.15132 | −0.14997 | −0.14167 | −0.13503 | 1.5 | 733.89 | 2 | 0.03 | 2 |
| −0.1468 | −0.1518 | −0.1468 | −0.14181 | −0.13781 | 1.5 | 733.89 | 2 | 0.02 | 1.33 |
| −0.13751 | −0.13751 | −0.13751 | −0.13751 | −0.13751 | 1.5 | 733.89 | 2 | 0.01 | 0.67 |
| −0.14023 | −0.14023 | −0.14023 | −0.14023 | −0.14023 | 1.501 | 545.61 | 2 | 0.01 | 0.67 |
| −0.13245 | −0.13245 | −0.13245 | −0.13245 | −0.13245 | 1.601 | 521.45 | 2 | 0.01 | 0.62 |
| −0.1379 | −0.1379 | −0.1379 | −0.1379 | −0.1379 | 1.601 | 521.45 | 2 | 0.01 | 0.62 |
| −0.16961 | −0.1781 | −0.15739 | −0.14814 | −0.1302 | 1.601 | 521.45 | 2 | 0.26 | 16.24 |
| −0.15795 | −0.16553 | −0.15394 | −0.14921 | −0.1341 | 1.601 | 521.45 | 2 | 0.13 | 8.12 |
| −0.21458 | −0.24177 | −0.22215 | −0.18569 | −0.13197 | 1.601 | 521.45 | 1 | 0.89 | 55.59 |
| −0.18542 | −0.21524 | −0.18213 | −0.15903 | −0.1423 | 1.635 | 554.6 | 2 | 0.25 | 15.29 |

| Average | 25 th Percentile | 50th Percentile | 75th Percentile | 95th Percentile | Area(ha) | Perimeter (m) | Clear-Cut | Area Trend (ha) | Relation Index Descent Area—Stand Area (%) |
|---|---|---|---|---|---|---|---|---|---|
| −0.14444 | −0.14444 | −0.14444 | −0.14444 | −0.14444 | 1.635 | 554.6 | 2 | 0.01 | 0.61 |
| −0.15267 | −0.15817 | −0.15136 | −0.14126 | −0.13561 | 1.635 | 554.6 | 2 | 0.06 | 3.67 |
| −0.21716 | −0.25869 | −0.20498 | −0.16668 | −0.13708 | 1.635 | 554.6 | 2 | 0.83 | 50.76 |
| −0.15287 | −0.15985 | −0.15117 | −0.14635 | −0.14072 | 1.635 | 554.6 | 2 | 0.05 | 3.06 |
| −0.21835 | −0.26563 | −0.22286 | −0.16413 | −0.1319 | 1.635 | 554.6 | 1 | 0.25 | 15.29 |
| −0.29059 | −0.33333 | −0.30054 | −0.26215 | −0.158 | 1.655 | 606.03 | 1 | 1.48 | 89.43 |
| −0.15938 | −0.17455 | −0.15357 | −0.14501 | −0.13479 | 1.655 | 606.03 | 2 | 0.06 | 3.63 |
| −0.15892 | −0.17544 | −0.15133 | −0.13647 | −0.13053 | 1.655 | 606.03 | 2 | 0.47 | 28.4 |
| −0.16947 | −0.16947 | −0.16947 | −0.16947 | −0.16947 | 1.667 | 507.11 | 2 | 0.01 | 0.6 |
| −0.14399 | −0.15113 | −0.14399 | −0.13685 | −0.13114 | 1.667 | 507.11 | 2 | 0.02 | 1.2 |
| −0.19546 | −0.24344 | −0.17814 | −0.14742 | −0.13883 | 1.667 | 507.11 | 2 | 0.06 | 3.6 |
| −0.14766 | −0.15024 | −0.14766 | −0.14507 | −0.143 | 1.723 | 678.92 | 2 | 0.02 | 1.16 |
| −0.1584 | −0.17022 | −0.14619 | −0.13219 | −0.12643 | 1.723 | 678.92 | 2 | 0.86 | 49.91 |
| −0.13257 | −0.13461 | −0.13288 | −0.13068 | −0.12892 | 1.723 | 678.92 | 2 | 0.03 | 1.74 |
| −0.13169 | −0.13169 | −0.13169 | −0.13169 | −0.13169 | 1.723 | 678.92 | 2 | 0.01 | 0.58 |
| −0.18098 | −0.18154 | −0.18098 | −0.18041 | −0.17996 | 1.723 | 678.92 | 2 | 0.02 | 1.16 |
| −0.14107 | −0.14878 | −0.12929 | −0.1279 | −0.12598 | 1.723 | 678.92 | 2 | 0.06 | 3.48 |
| −0.21704 | −0.24281 | −0.21843 | −0.19391 | −0.13694 | 1.723 | 678.92 | 1 | 1.63 | 94.6 |
| −0.14814 | −0.14814 | −0.14814 | −0.14814 | −0.14814 | 1.724 | 506.48 | 2 | 0.01 | 0.58 |
| −0.13965 | −0.13965 | −0.13965 | −0.13965 | −0.13965 | 1.724 | 506.48 | 2 | 0.01 | 0.58 |
| −0.35562 | −0.41459 | −0.37857 | −0.32367 | −0.17834 | 1.724 | 506.48 | 1 | 1.57 | 91.07 |
| −0.12671 | −0.12671 | −0.12671 | −0.12671 | −0.12671 | 1.724 | 506.48 | 2 | 0.01 | 0.58 |
| −0.13364 | −0.13364 | −0.13364 | −0.13364 | −0.13364 | 1.724 | 506.48 | 2 | 0.01 | 0.58 |
| −0.1421 | −0.14817 | −0.14124 | −0.13316 | −0.12655 | 1.724 | 506.48 | 2 | 0.65 | 37.7 |
| −0.25479 | −0.30969 | −0.25084 | −0.18642 | −0.14496 | 1.766 | 953.43 | 1 | 1.14 | 64.55 |
| −0.23973 | −0.2695 | −0.24759 | −0.21446 | −0.17215 | 1.812 | 623.92 | 1 | 0.43 | 23.73 |
| −0.16133 | −0.16133 | −0.16133 | −0.16133 | −0.16133 | 1.812 | 623.92 | 2 | 0.01 | 0.55 |
| −0.14372 | −0.15016 | −0.14372 | −0.13729 | −0.13214 | 1.812 | 623.92 | 2 | 0.02 | 1.1 |
| −0.27356 | −0.34141 | −0.2691 | −0.20169 | −0.14752 | 1.829 | 579.64 | 2 | 0.58 | 31.71 |
| −0.15457 | −0.16998 | −0.13838 | −0.12976 | −0.12683 | 1.829 | 579.64 | 2 | 0.14 | 7.65 |
| −0.21968 | −0.25895 | −0.21919 | −0.17942 | −0.13387 | 1.829 | 579.64 | 1 | 1.16 | 63.42 |
| −0.13046 | −0.13046 | −0.13046 | −0.13046 | −0.13046 | 1.831 | 571.67 | 2 | 0.01 | 0.55 |
| −0.14231 | −0.14954 | −0.14633 | −0.1371 | −0.12971 | 1.831 | 571.67 | 2 | 0.03 | 1.64 |
| −0.27859 | −0.39149 | −0.24764 | −0.1722 | −0.14462 | 1.831 | 571.67 | 1 | 1.51 | 82.47 |
| −0.17359 | −0.20475 | −0.17514 | −0.1361 | −0.1282 | 1.831 | 571.67 | 2 | 0.18 | 9.83 |
| −0.14677 | −0.14741 | −0.14677 | −0.14613 | −0.14562 | 1.855 | 624.87 | 2 | 0.02 | 1.08 |
| −0.23404 | −0.28065 | −0.23151 | −0.17762 | −0.13931 | 1.855 | 624.87 | 1 | 1.24 | 66.85 |
| −0.23753 | −0.26188 | −0.24376 | −0.22396 | −0.155 | 1.879 | 600.57 | 1 | 1.61 | 85.68 |
| −0.13107 | −0.13107 | −0.13107 | −0.13107 | −0.13107 | 1.879 | 600.57 | 2 | 0.01 | 0.53 |
| −0.1435 | −0.14644 | −0.13951 | −0.13263 | −0.13081 | 1.888 | 602.28 | 2 | 0.11 | 5.83 |
| −0.13331 | −0.1358 | −0.13331 | −0.13081 | −0.12882 | 1.888 | 602.28 | 2 | 0.02 | 1.06 |
| −0.40585 | −0.45969 | −0.42615 | −0.38028 | −0.21548 | 1.899 | 602.45 | 1 | 1.85 | 97.42 |
| −0.17522 | −0.21841 | −0.15497 | −0.13357 | −0.12836 | 1.899 | 602.45 | 2 | 0.21 | 11.06 |
| −0.14316 | −0.14316 | −0.14316 | −0.14316 | −0.14316 | 1.899 | 602.45 | 2 | 0.01 | 0.53 |

| Average | 25 th Percentile | 50th Percentile | 75th Percentile | 95th Percentile | Area(ha) | Perimeter (m) | Clear-Cut | Area Trend (ha) | Relation Index Descent Area—Stand Area (%) |
|---|---|---|---|---|---|---|---|---|---|
| −0.22906 | −0.24988 | −0.21584 | −0.19501 | −0.15383 | 1.899 | 602.45 | 2 | 0.04 | 2.11 |
| −0.13006 | −0.13006 | −0.13006 | −0.13006 | −0.13006 | 1.899 | 602.45 | 2 | 0.01 | 0.53 |
| −0.21874 | −0.23649 | −0.20481 | −0.1736 | −0.14765 | 1.903 | 593.22 | 2 | 0.3 | 15.76 |
| −0.26504 | −0.29645 | −0.27324 | −0.23117 | −0.15774 | 1.903 | 593.22 | 1 | 1.56 | 81.98 |
| −0.16172 | −0.16642 | −0.16172 | −0.15701 | −0.15325 | 1.903 | 593.22 | 2 | 0.02 | 1.05 |
| −0.15651 | −0.17117 | −0.15099 | −0.13909 | −0.12957 | 1.903 | 593.22 | 2 | 0.03 | 1.58 |
| −0.14408 | −0.14408 | −0.14408 | −0.14408 | −0.14408 | 1.903 | 593.22 | 2 | 0.01 | 0.53 |
| −0.2802 | −0.31621 | −0.28764 | −0.26531 | −0.15291 | 1.905 | 1208.87 | 2 | 0.43 | 22.57 |
| −0.21117 | −0.25255 | −0.18013 | −0.12749 | −0.12665 | 1.905 | 1208.87 | 2 | 0.05 | 2.62 |
| −0.16839 | −0.18316 | −0.16214 | −0.14968 | −0.14226 | 1.918 | 580.19 | 2 | 0.07 | 3.65 |
| −0.14233 | −0.14324 | −0.13267 | −0.13176 | −0.13116 | 1.918 | 580.19 | 2 | 0.04 | 2.09 |
| −0.35707 | −0.41476 | −0.37546 | −0.325 | −0.20332 | 1.918 | 580.19 | 1 | 1.8 | 93.85 |
| −0.14677 | −0.16064 | −0.1368 | −0.12905 | −0.12666 | 1.918 | 580.19 | 2 | 0.4 | 20.86 |
| −0.42203 | −0.49934 | −0.45359 | −0.36269 | −0.20965 | 1.928 | 559.54 | 1 | 1.87 | 96.99 |
| −0.14536 | −0.16062 | −0.13847 | −0.13189 | −0.12627 | 1.928 | 559.54 | 2 | 0.14 | 7.26 |
| −0.19943 | −0.22316 | −0.18342 | −0.16597 | −0.13969 | 1.928 | 559.54 | 2 | 0.25 | 12.97 |
| −0.16497 | −0.17498 | −0.15693 | −0.1439 | −0.12733 | 1.928 | 559.54 | 2 | 0.17 | 8.82 |
| −0.44469 | −0.50894 | −0.48239 | −0.43041 | −0.22805 | 1.998 | 861.4 | 1 | 1.44 | 72.07 |
| −0.16154 | −0.16812 | −0.16241 | −0.15582 | −0.14541 | 1.998 | 861.4 | 2 | 0.04 | 2 |
| −0.15513 | −0.17241 | −0.15292 | −0.13564 | −0.1351 | 1.998 | 861.4 | 2 | 0.04 | 2 |
| −0.17521 | −0.19718 | −0.19469 | −0.16298 | −0.13761 | 1.998 | 861.4 | 2 | 0.03 | 1.5 |
| −0.17475 | −0.18091 | −0.17802 | −0.14946 | −0.14795 | 1.998 | 861.4 | 2 | 0.05 | 2.5 |
| −0.15726 | −0.1712 | −0.14467 | −0.13782 | −0.1322 | 1.998 | 861.4 | 2 | 0.07 | 3.5 |
| −0.16625 | −0.16668 | −0.16625 | −0.16583 | −0.16549 | 1.998 | 861.4 | 2 | 0.02 | 1 |
| −0.14579 | −0.14579 | −0.14579 | −0.14579 | −0.14579 | 2.041 | 1006.91 | 2 | 0.01 | 0.49 |
| −0.40849 | −0.4701 | −0.44054 | −0.37299 | −0.217 | 2.041 | 1006.91 | 1 | 1.07 | 52.43 |
| −0.27048 | −0.35237 | −0.22803 | −0.19202 | −0.15608 | 2.041 | 1006.91 | 2 | 0.17 | 8.33 |
| −0.14808 | −0.14808 | −0.14808 | −0.14808 | −0.14808 | 2.073 | 679.3 | 2 | 0.01 | 0.48 |
| −0.16697 | −0.19234 | −0.15237 | −0.12856 | −0.12753 | 2.073 | 679.3 | 2 | 0.05 | 2.41 |
| −0.16541 | −0.17958 | −0.15549 | −0.1442 | −0.13403 | 2.129 | 681.24 | 2 | 0.14 | 6.58 |
| −0.14749 | −0.15472 | −0.13785 | −0.13626 | −0.12714 | 2.129 | 681.24 | 2 | 0.09 | 4.23 |
| −0.14746 | −0.15549 | −0.14587 | −0.13875 | −0.13161 | 2.129 | 681.24 | 2 | 0.08 | 3.76 |
| −0.26944 | −0.33451 | −0.26908 | −0.20023 | −0.14922 | 2.129 | 681.24 | 1 | 1.15 | 54.02 |
| −0.15199 | −0.15244 | −0.15101 | −0.13573 | −0.13368 | 2.186 | 714.93 | 2 | 0.05 | 2.29 |
| −0.19907 | −0.22991 | −0.1981 | −0.16397 | −0.14258 | 2.186 | 714.93 | 1 | 0.91 | 41.63 |
| −0.18884 | −0.20484 | −0.17036 | −0.16146 | −0.13573 | 2.186 | 714.93 | 2 | 0.19 | 8.69 |
| −0.16786 | −0.17711 | −0.16785 | −0.14982 | −0.12863 | 2.31 | 633.77 | 2 | 0.2 | 8.66 |
| −0.27451 | −0.32659 | −0.28688 | −0.22165 | −0.14687 | 2.31 | 633.77 | 1 | 1.47 | 63.64 |
| −0.13756 | −0.13756 | −0.13756 | −0.13756 | −0.13756 | 2.31 | 633.77 | 2 | 0.01 | 0.43 |
| −0.16142 | −0.15187 | −0.14183 | −0.1306 | −0.12683 | 2.31 | 633.77 | 2 | 0.13 | 5.63 |
| −0.16957 | −0.19903 | −0.15774 | −0.14356 | −0.12758 | 2.31 | 633.77 | 2 | 0.27 | 11.69 |
| −0.12854 | −0.12854 | −0.12854 | −0.12854 | −0.12854 | 2.31 | 633.77 | 2 | 0.01 | 0.43 |
| −0.1666 | −0.1937 | −0.16034 | −0.13962 | −0.12763 | 2.31 | 633.77 | 2 | 0.07 | 3.03 |
| −0.14834 | −0.17271 | −0.13304 | −0.13226 | −0.13023 | 2.31 | 633.77 | 2 | 0.05 | 2.16 |

| Average | 25 th Percentile | 50th Percentile | 75th Percentile | 95th Percentile | Area(ha) | Perimeter (m) | Clear-Cut | Area Trend (ha) | Relation Index Descent Area—Stand Area (%) |
|---|---|---|---|---|---|---|---|---|---|
| −0.27967 | −0.34401 | −0.2738 | −0.21573 | −0.15057 | 2.322 | 786.48 | 1 | 1.74 | 74.94 |
| −0.58509 | −0.7265 | −0.50911 | −0.42466 | −0.39463 | 2.322 | 786.48 | 2 | 0.22 | 9.47 |
| −0.14111 | −0.15155 | −0.14154 | −0.1311 | −0.12844 | 2.322 | 786.48 | 2 | 0.04 | 1.72 |
| −0.13559 | −0.13559 | −0.13559 | −0.13559 | −0.13559 | 2.359 | 642.38 | 2 | 0.01 | 0.42 |
| −0.13421 | −0.13715 | −0.13421 | −0.13126 | −0.12891 | 2.359 | 642.38 | 2 | 0.02 | 0.85 |
| −0.22584 | −0.27229 | −0.239 | −0.18401 | −0.13344 | 2.37 | 951.94 | 2 | 0.11 | 4.64 |
| −0.29159 | −0.34434 | −0.31059 | −0.26154 | −0.15669 | 2.37 | 951.94 | 1 | 1.13 | 47.68 |
| −0.17862 | −0.2052 | −0.17862 | −0.15646 | −0.13212 | 2.37 | 951.94 | 2 | 0.24 | 10.13 |
| −0.24756 | −0.25748 | −0.19063 | −0.15816 | −0.13316 | 2.37 | 951.94 | 2 | 0.77 | 32.49 |
| −0.44019 | −0.57204 | −0.48096 | −0.33548 | −0.18443 | 2.37 | 951.94 | 2 | 0.19 | 8.02 |
| −0.17063 | −0.18597 | −0.15568 | −0.13804 | −0.12799 | 2.37 | 951.94 | 2 | 0.29 | 12.24 |
| −0.13837 | −0.13776 | −0.13123 | −0.12571 | −0.12552 | 2.409 | 747.21 | 2 | 0.08 | 3.32 |
| −0.22174 | −0.25322 | −0.223 | −0.18728 | −0.14233 | 2.409 | 747.21 | 1 | 1.77 | 73.47 |
| −0.14537 | −0.14568 | −0.14183 | −0.13479 | −0.12695 | 2.409 | 747.21 | 2 | 0.12 | 4.98 |
| −0.21647 | −0.25071 | −0.2202 | −0.18282 | −0.14216 | 2.412 | 665.46 | 1 | 1.7 | 70.48 |
| −0.15635 | −0.17378 | −0.146 | −0.13412 | −0.12755 | 2.412 | 665.46 | 2 | 0.31 | 12.85 |
| −0.13728 | −0.13546 | −0.13494 | −0.13319 | −0.12875 | 2.412 | 665.46 | 2 | 0.09 | 3.73 |
| −0.16545 | −0.178 | −0.14808 | −0.14422 | −0.14112 | 2.412 | 665.46 | 2 | 0.03 | 1.24 |
| −0.17012 | −0.20199 | −0.1734 | −0.14153 | −0.1319 | 2.447 | 880.22 | 2 | 0.04 | 1.63 |
| −0.26061 | −0.31428 | −0.25081 | −0.19218 | −0.14863 | 2.453 | 955.81 | 1 | 0.6 | 24.46 |
| −0.15961 | −0.17172 | −0.1535 | −0.1461 | −0.13205 | 2.453 | 955.81 | 2 | 0.24 | 9.78 |
| −0.21687 | −0.26153 | −0.21099 | −0.17286 | −0.13568 | 2.53 | 1094.23 | 2 | 0.51 | 20.16 |
| −0.15715 | −0.15715 | −0.15715 | −0.15715 | −0.15715 | 2.53 | 1094.23 | 2 | 0.01 | 0.4 |
| −0.16631 | −0.18626 | −0.17947 | −0.15295 | −0.13173 | 2.53 | 1094.23 | 2 | 0.03 | 1.19 |
| −0.23432 | −0.28654 | −0.23014 | −0.18786 | −0.1423 | 2.53 | 1094.23 | 1 | 0.22 | 8.7 |
| −0.40136 | −0.4617 | −0.41943 | −0.38694 | −0.20181 | 2.558 | 698.54 | 1 | 1.49 | 58.25 |
| −0.42643 | −0.48943 | −0.46376 | −0.39539 | −0.21007 | 2.558 | 698.54 | 2 | 0.49 | 19.16 |
| −0.38888 | −0.44878 | −0.41479 | −0.34925 | −0.20973 | 2.67 | 819.01 | 1 | 2.62 | 98.13 |
| −0.12747 | −0.12747 | −0.12747 | −0.12747 | −0.12747 | 2.742 | 1049.34 | 2 | 0.01 | 0.36 |
| −0.25849 | −0.33711 | −0.23941 | −0.16826 | −0.13405 | 2.742 | 1049.34 | 1 | 0.63 | 22.98 |
| −0.16405 | −0.19545 | −0.15011 | −0.13947 | −0.12741 | 2.742 | 1049.34 | 2 | 0.25 | 9.12 |
| −0.1743 | −0.20203 | −0.16356 | −0.14341 | −0.13001 | 2.742 | 1049.34 | 2 | 0.64 | 23.34 |
| −0.23987 | −0.31878 | −0.23027 | −0.15391 | −0.12828 | 2.742 | 1049.34 | 2 | 1.09 | 39.75 |
| −0.13797 | −0.14491 | −0.13457 | −0.12854 | −0.12828 | 2.742 | 1049.34 | 2 | 0.05 | 1.82 |
| −0.38668 | −0.42222 | −0.39961 | −0.37636 | −0.24609 | 2.762 | 695.91 | 1 | 2.64 | 95.58 |
| −0.14931 | −0.13605 | −0.13122 | −0.13046 | −0.12889 | 2.762 | 695.91 | 2 | 0.05 | 1.81 |
| −0.14088 | −0.14327 | −0.13986 | −0.1329 | −0.13015 | 2.762 | 695.91 | 2 | 0.11 | 3.98 |
| −0.17711 | −0.19887 | −0.1702 | −0.15379 | −0.1402 | 2.762 | 695.91 | 2 | 0.06 | 2.17 |
| −0.14731 | −0.15537 | −0.14337 | −0.13728 | −0.13241 | 2.762 | 695.91 | 2 | 0.03 | 1.09 |
| −0.17349 | −0.18653 | −0.16118 | −0.14802 | −0.13844 | 2.763 | 848.14 | 2 | 0.07 | 2.53 |
| −0.3288 | −0.39537 | −0.34111 | −0.26649 | −0.16446 | 2.763 | 848.14 | 1 | 2.31 | 83.6 |
| −0.17916 | −0.17916 | −0.17916 | −0.17916 | −0.17916 | 2.801 | 688.38 | 2 | 0.01 | 0.36 |
| −0.13097 | −0.13097 | −0.13097 | −0.13097 | −0.13097 | 2.801 | 688.38 | 2 | 0.01 | 0.36 |
| −0.40575 | −0.47278 | −0.42793 | −0.36401 | −0.21225 | 2.801 | 688.38 | 1 | 2.63 | 93.9 |

| Average | 25 th Percentile | 50th Percentile | 75th Percentile | 95th Percentile | Area(ha) | Perimeter (m) | Clear-Cut | Area Trend (ha) | Relation Index Descent Area—Stand Area (%) |
|---------|------------------|-----------------|-----------------|-----------------|----------|---------------|-----------|-----------------|---------------------------------------------|
| −0.16495 | −0.16495 | −0.16495 | −0.16495 | −0.16495 | 2.801 | 688.38 | 2 | 0.01 | 0.36 |
| −0.23812 | −0.28166 | −0.22923 | −0.17898 | −0.14735 | 2.801 | 688.38 | 2 | 0.09 | 3.21 |
| −0.14572 | −0.15132 | −0.1406 | −0.13756 | −0.13513 | 2.927 | 838.93 | 2 | 0.03 | 1.02 |
| −0.16722 | −0.17326 | −0.15136 | −0.13978 | −0.13519 | 2.927 | 838.93 | 2 | 0.09 | 3.07 |
| −0.38807 | −0.44128 | −0.42472 | −0.36957 | −0.15597 | 2.927 | 838.93 | 1 | 2.69 | 91.9 |
| −0.18853 | −0.18853 | −0.18853 | −0.18853 | −0.18853 | 2.927 | 838.93 | 2 | 0.01 | 0.34 |

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
