# Peer review of "Multi-Temporal Sentinel-2 Data Analysis for Smallholding Forest Cut Control"

_remotesensing, doi:10.3390/rs13152983_

Round 1
Reviewer 1 Report
- line 46-85: This study is focused on the possibility of detecting forest management activities (cutting/reforestation) in a small area using satellite images, so the contents related to GVA and forest management certification(FSC/PEFC) need to be written briefly.
- line 128-130: Checking of the area unit (ex, 2,040,754 million ha)
- Figure 5 (a) / Figure 6 (a): The low resolution of the figure makes the information unreadable.
- Figure 6 (b): Missing legend information about solid lines and dots
- Line 378: Check figure number (Figure 4.a → Figure 5.a or 5.b)
- Line 386-389: Check figure number (ex, Figure 5.a → Figure 7.a)
- Line 403-405: Check figure number
- Table 2: Each statistical indicator including equation and description be explained in the methods.
- Line 552 and 557-558: Typing mistake(NVDI → NDVI)
- Overall: It will be helpful for readers to understand how the multi-temporal NDVI was used, and how the changes inter-annual variation were taken into account are described in the results.
- Reference: References must be modified according to MPI guidelines (names of all authors, etc.) (ex, line 596)
Author Response
Dear reviewer 1,
Please see the attachment
Reviewer 2 Report
This paper proposed a method to detect the traits of cutting in small plots of forest based on high resolution data of Sentinel-2 NDVI. This study is a typical one of the applications of using newly launched satellite product. Meanwhile, the quick change of forest including the felling and extreme disasters has been hot points for relative research in the past decades. However, after reading the whole paper, I still felt a little bit confused of the main implication of this study. For example, it seemed that the emphasis of this paper is to propose the new approach, but the relevant methods were not clearly introduced. As you may know, there has been a lot of ways for detecting the drop of tree growing or logging in the past, from Sentinel 2 (Please check their titles in the last). Honestly, I did not see the novelty of this method raised in this paper. If you want to highlight the importance and novelty of your method, some necessary comparisons with other methods and strict evaluation should be carried on. On the other hand, I strongly recommend the authors to improve the introduction of the approach in this paper. For instance, there are a lot of mismatches of Figures in the Section of Result, which resulted in many confusions for understanding when reading these passages. Some other details like how to define the threshold for detection were also not clearly explained. Some other points that I have reviewed were shown below.
- Line 37. The format of “100 g m−2 a−1” should be corrected.
- Paragraph 2. Please unify the marks of “euros” and “€”.
- Line 51-61. The two papers need to be simplified into one or two sentences. In fact, the whole introduction is too long (nine paragraphs totally) to catch the main point of this paper. For example, the passage of Line 63 does not have much to do with the purpose of this study, which can be greatly shortened or even deleted. 3-4 Paragraphs are enough for the Introduction in this study, I suppose.
- Please declare the coordinated system in this map.
- Line 180. What is the difference between small parcels and plots?
- Line 204. How did you extract the true classification of each object, which also acted as validation data in the later parts.
- Line 220. Please clarify the aim of change the value from 5% to 15%.
- Line 248. What is the “several visual tests”? How to choose the threshold?
- Figure 3, why did you choose the value of -0.1254? Does this value refer to the mean difference of all the periods or the difference between two periods? In addition, I guess you mix the usage of “trend” and “difference”, of which the former one represents the long-term change while the latter one denotes the variation between two time points.
- The flowchart should be improved.
- Line 324, how could we know that the method successfully detected the sudden change or not?
- Line 339, should be “spatial and temporal”.
- Figure 5b, is this line belong to globulus or the mean value of globulus with meadows and crops?
- Figure 7a and Figure 7b lack the x/y labels.
- Line 384. Should all the Figures in this paragraph be Figure 7? With the next paragraph being Figure 8?
- Line 453-461. This part is quite confusing. Does the cloudness influence the detection or not? The first sentence showed “Sentinel-2 is vulnerable to cloudy weather”, but the latter ones indicated it “could be considered negligible”, and later, you still use the threshold to “reduce the noise caused by cloudiness,”
- In the conclusion, I did not catch the advantage of the method you raised in this paper. In the sentence of Line 567, the authors have mentioned it can “reduce the processing time and storage capacity.”, but I did not find the proof of this argument throughout the whole paper.
Some other questions I had after reading are:
- Why you did not introduce the result of the whole study region as shown in Figure 1, but only show several plots? How about other regions?
- How to test the accuracy of the classification?
- The authors also mentioned “the size and shape of the plots were also verified to affect the accuracy, as did the presence of uncut deciduous trees or simultaneous agricultural uses” in the abstract, but this conclusion was not well supported or proved, just like a common sense.
References
- Detection And Characterization of Forest Harvesting In Piedmont Through Sentinel-2 Imagery: A Methodological Proposal
- Detecting Harvest Events in Plantation Forest Using Sentinel-1 and -2 Data via Google Earth Engine.
- Sentinel-2 Imagery Processing for Tree Logging Observations on the Białowieża Forest World Heritage Site
Author Response
Dear reviewer 2,
Please see the attachment

Reviewer 3 Report
The study focused on multi-temporal analysis of Sentinel-2 data for smallholding forest cut control
comments:
- Why did not you consider land-use change monitoring to get to know the change between the forests and other land-use categories such as agriculture or other categories?
- What is the spatial resolution of Sentinel-2 data used in the current study?
- If you performed multitemporal classification as mentioned, which classifiers were used and why? and which variables were used in the classification? clarify, and provide more information about the classification process.
- change in NDVI might be caused by different factors including seasonality change, forest fire, forest transformation to other land-use categories, and forest cutting.
Author Response
Dear reviewer 3,
Please see the attachment

Round 2
Reviewer 1 Report
- Why did you apply 14 statistical indicators in table 2? If evaluation of all indicators is required, you have to interpret the analysis results for each indicator (table 3) in the results.
- Line 106-108: Checking the unit of 2,040 million ha and 1,500 million ha (?)
- line 308, line 486-489, line 522: NVDI and NDVI index -> NDVI
Author Response
Dear Reviewer 1,
Please see the attachment

Reviewer 2 Report
In this version, the authors have greatly improved the introduction and expression of the paper. However, I still feel concerned about the novelty of this method the authors used. For example, the key point of the application is the threshold of NDVI change, which is too common and too simple for ground-object recognition in remote sensing. If you checked more papers studying this kind of issue, there are a lot of different new approaches which have significantly enhanced the accuracy of the detection, like importing artificial neural networks, random forests, bayesian algorithm, based on not only multi-bands information, but also texture and structure of the objective.
Meanwhile, just as I mentioned in the first review, Figure 1 should include the plots that you have tested but exclude all the others that you haven't finished.
Author Response
Dear Reviewer 2,
Please see the attachment

Reviewer 3 Report
Accept in present form.
Author Response
Dear Reviewer 3,
Thank you very much for your final assessment. We are grateful for all your comments, as well as for every suggestion. We appreciate your contribution in improving this manuscript. We have an English revision made by native speakers (see attached).
Best regards,
the authors
